# BOptformer: Beyond Transformer for Black-box Optimization

## Abstract

We design a novel Transformer for continuous unconstrained black-box optimization, called BOptformer. Inspired by the similarity between Vision Transformer and evolutionary algorithms (EAs), we modify Tansformer's multi-head self-attention layer, feed-forward network, and residual connection to implement the functions of crossover, mutation, and selection operators. Moreover, we devise an iterated mode to generate and survive potential solutions like EAs. BOptformer learns the optimization strategies from the target task automatically without human intervention, which addresses the poor generalization of human-designed EAs when given a new task. Compared to baselines, such as EAs, Bayesian optimization, and the learning-to-optimize (L2O) method, BOptformer shows the top performance in six black-box functions and two real-world applications. We also find that untrained BOptformer can achieve good performance on the simple tasks. Deep BOptformer performs better than shallow ones. We bring a new and efficient Transformer-based black-box optimization framework for the L2O and EA communities.

## 1 Introduction

Many tasks, such as neural architecture search (Elsken et al., 2019) and hyperparameter optimization (Hutter et al., 2019; Golovin et al., 2017), can be abstracted as black-box optimization problems, which means that although we can evaluate $f(x)$ for any $x \in X$, we have no access to any other information about $f$, such as the Hessian and gradients. A series of hand-designed algorithms, such as evolutionary algorithms (EAs) (Mitchell, 1998; Khadka & Tumer, 2018; Zhang & Li, 2007), Bayesian optimization (Snoek et al., 2012; Mutny & Krause, 2018; Li et al., 2017; Kandasamy et al., 2015; Balandat et al., 2020), and evolutionary strategies (ES) (Wierstra et al., 2014; Hansen & Ostermeier, 2001; Auger & Hansen, 2005; Salimans et al., 2017), have been designed to solve black-box optimization.

Recently, the learning to optimize (L2O) framework (Chen et al., 2022) gives an new insight on optimization by leveraging the recurrent neural network (RNN), long short-term memory architecture (LSTM) (Chen et al., 2020; Andrychowicz et al., 2016; Chen et al., 2017; Li & Malik, 2016; Wichrowska et al., 2017; Bello et al., 2017) or multilayer perceptron (MLP) (Metz et al., 2019) as the optimizer to develop optimization methods, aiming at reducing the laborious iterations of hand engineering (Sun et al., 2018; Vicol et al., 2021; Flennerhag et al., 2021; Li & Malik, 2016; Sun et al., 2018). They don't concentrate on issues with black-box optimization. The core of L2O is constructing a strong mapping from the initial solutions to the optimal solution. Although several efforts like (Cao et al., 2019; Chen et al., 2017) have coped with the black-box problems, their effectiveness may be hindered by the limited representational capabilities of RNN, LSTM, and MLP.

In EAs, the hand-designed crossover, mutation, and selection operators make the initial population move near the optimal solution. This updated model has stood the test of time. Because the evolutionary operators must be modified to maximize their performance on the target task, human-designed EAs have a low generalization ability to a new black-box problem. Most notably, the limited use of target function information in EA design due to expert knowledge limitations makes it difficult to adapt to the target task. Learning the optimization strategies from the taget task is the key step to overcome this limitation.

This paper designs a novel L2O framework based on the advantages of Vision Transformer (Dosovitskiy et al., 2021) and EAs to overcome the above limitations, termed BOptformer. Moreover,

Transformer (Han et al., 2022) owns a strong representation ability, and there is currently no work to use Transformer for optimization. Inspired by the similarity of EAs and Transformer (Zhang et al., 2021; 2022), BOptformer revised the critical part of Transformer to realize the mapping from the random and optimal populations. To generate potential individuals to approach the optimal solution, we first design an self-attention (SA)-based crossover module (SAC) to simulate the crossover operator of EA, and then the output of this module is input into the proposed feed-forward network (FFN)-based mutation module (FM) to perform mutation. Moreover, the residual and selection module (RSSM) is designed to survive the fittest individuals. RSSM is a pairwise comparison between the output of SAC, FM, and the input population regarding their fitness. We design an BOptformer Block (OB) consisting of SAC, FM, and RSSM. Finally, we construct BOptformer by stacking OBs to simulate generations of EAs.

Moreover, to cope with black-box optimization, we establish a function set to train BOptformer under an unsupervised mode. We construct a set of differentiable functions with similar properties to the targeted black-box optimization problems. This training set contains the pair of the initial population and the designed function. Thus, we can use gradient-based methods to train BOptformer.

We tested BOptformer on six standard functions, the protein docking (Cao & Shen, 2020) problem, and the planar mechanic arm problem (Wang et al., 2021). The experimental results demonstrate the top rank of BOptformer and the strong representation compared with three population-based baselines, Bayesian optimization, and one learning-to-optimize method (Cao et al., 2019). Moreover, we also analyze the effect of learning rate, deep structure, and weight sharing between OBs. The highlights of this paper are summarized as follows:

1) We propose a solid Transformer-based L2O framework addressing black-box problems to the L2O community. We have demonstrated its benefit when compared with standard black-box optimization methods, particularly for the L2O-based method.

2) BOptformer efficiently uses the target black-box function's information to aid in the development of the optimization strategy. Compared to the human-designed EA, BOptformer has a substantially greater degree of task fit.

## 2  RELATED WORK

**Transformer**    Transformer structure achieves significant progress for machine translation task (Vaswani et al., 2017), computer vision task (Dosovitskiy et al., 2021), time series task (Zhou et al., 2021), and so on. Many improved models are proposed and obtain great achievements (Han et al., 2022). There are no Transformer-based efforts for handling optimization problems, which is crucial in the machine learning community. (Vaswani et al., 2017) proposed the meta-learning hyperparameter optimization framework with Transformers to learn both policy and function priors from data across different search spaces. However, the BOptformer proposed in this paper expands the application scope of Transformer and can effectively deal with this case. The basic modules of Transformer are shown in Appendix A.1.

**Evolutionary Algorithm**    Inspired by the evolution of species, EAs have provided surprising performance for black-box optimization (Mitchell, 1998). The basic modules of EAs are shown in Appendix A.2. Many influential variants have been proposed to deal with different problems (Das & Suganthan, 2010; Wu & Liu, 2019), but at their core they are: 1) recombination and mutation, how to produce the excellent solution; 2) selection, how to choose the best individuals between the parents and offspring. Thus, many algorithmic components have been designed for different tasks. The performance of algorithms varies towards various tasks, as different optimization strategies may be required given diverse landscapes. Current methods manually adjust genetic operators' hyperparameters and design the combination between them (Kerschke et al., 2019; Tian et al., 2020) to map the random population to the optimal solution. We require an expert to design or choose the evolutionary operations when given a new black-box optimization task to maximize its performance on the target task, which negatively impacts generalization ability. Most notably, the limited use of target function information in EA design due to expert knowledge limitations makes it difficult to adapt to the target task. The suggested BOptformer uses a Transformer framework instead of the manually designed crossover, mutation, and selection operators. The genetic operator is then designed automatically by the built Transformer rather than by a human designer. BOptformer efficiently

uses the target black-box function's information to aid in developing the optimization strategy. In comparison to the human-designed EA, BOptformer has a substantially greater degree of task fit.

## 3 BOPTFORMER

### 3.1 PROBLEM DEFINITION

A black-box optimization problem can be transformed as a minimization problem, as shown in Equation (1), and constraints may exist for corresponding solutions:

$$\min \ f(\boldsymbol{x}), s.t. \ x_i \in [l_i, u_i] \tag{1}$$

where $\boldsymbol{x} = (x_1, x_2, \cdots, x_d)$ represents the solution of optimization problem $f$, the lower and upper bounds $\boldsymbol{l} = (l_1, l_2, \cdots, l_d)$ and $\boldsymbol{u} = (u_1, u_2, \cdots, u_d)$, and $d$ is the dimension of $\boldsymbol{x}$. Suppose $n$ individuals of one population be $\boldsymbol{X}_1 = (X_{1,1}, X_{1,2}, \cdots, X_{1,d}), \boldsymbol{X}_2 = (X_{2,1}, X_{2,2}, \cdots, X_{2,d}), \cdots, \boldsymbol{X}_n = (X_{n,1}, X_{n,2}, \cdots, X_{n,d})$, then BOptformer are required to find the population near the optimal solution $\hat{\boldsymbol{x}}$. We suppose that $\boldsymbol{X}^0$ is the initial population and $\boldsymbol{X}^t$ is the output population.

### 3.2 SELF-ATTENTION CROSSOVER MODULE

Similar to the crossover operator in EAs, we propose a new module based on SA to generate the potential solutions by maximizing information interaction among individuals in a population. The crossover operator generates a new individual by $\sum_{i=1}^{n} \boldsymbol{X}_i \boldsymbol{W}_i^c$ (Zhang et al., 2021). $\boldsymbol{W}_i^c$ is the diagonal matrix. If $\boldsymbol{W}_i^c$ is full of zeros, the $i$th individual has no contribution. Suppose a population $\boldsymbol{X}$ is arranged in a non-descending order of fitness, and $\boldsymbol{F} \in \mathbb{R}^{n \times 1}$ be the fitness matrix of $\boldsymbol{X}$. Then, this module can be represented as follows:

$$\boldsymbol{X}^c = SAC(\boldsymbol{X}, \boldsymbol{F}) \tag{2}$$

where $\boldsymbol{X}^c$ is the output population of the proposed SAC module.

Since the object processed by BOptformer is the population, and the order of individuals in the population does not affect the population distribution, SA does not require position coding. Standard SA projects the input sequence $\boldsymbol{X}$ into a $d$-dimensional space via the queries ($\boldsymbol{Q}$), keys ($\boldsymbol{K}$), and values ($\boldsymbol{V}$). These three mappings enable the SA module to capture better the characteristics of the problems encountered during training. In other words, these three mappings strengthen the ability of SA to focus on specific problems but do not necessarily make SA have good transferability between different problems. Therefore, we consider removing these three mappings for enhanced transferability, and $\boldsymbol{X}^c = \boldsymbol{A}\boldsymbol{X}$. $\boldsymbol{A} \in \mathbb{R}^{n \times n}$ is a self-attention matrix that can be learned to maximize inter-individual information interaction based on individual ranking information. This is why the population needs to be sorted in non-descending order.

However, designing crossover operations based solely on population ranking information is a coarse-grained approach. Because this method only considers the location information of individuals in the population, but does not consider the fitness relationship between individuals. Therefore, we further introduce fitness information to assist in learning crossover operators:

$$\boldsymbol{A}^F = SA(\boldsymbol{F}) = Softmax\left(\boldsymbol{F}\boldsymbol{W}^Q(\boldsymbol{F}\boldsymbol{W}^K)^T / sqrt(d_k)\right)$$

Thus, $\boldsymbol{X}^c = \boldsymbol{A}\boldsymbol{X} + \boldsymbol{A}^F\boldsymbol{X}$. To better balance the roles of $\boldsymbol{A}$ and $\boldsymbol{A}^F$, we introduce two learnable weights $\boldsymbol{W}_1^c \in \mathbb{R}^{n \times 1}$ and $\boldsymbol{W}_2^c \in \mathbb{R}^{n \times 1}$. Therefore, the final crossover operation is shown as follows:

$$\boldsymbol{X}^c = tile(\boldsymbol{W}_1^c) \odot (\boldsymbol{A}\boldsymbol{X}) + tile(\boldsymbol{W}_2^c) \odot (\boldsymbol{A}^F\boldsymbol{X}) \tag{3}$$

where $\boldsymbol{X}^c \in \mathbb{R}^{n \times d}$ is the population obtained by $\boldsymbol{X}$ through the SAC module; $\odot$ represents Hadamard product; the *tile* copy function extends the vector to a matrix.

### 3.3 FFN-BASED MUTATION MODULE

The mutation operator brings random changes into the population. Specifically, an individual $\boldsymbol{X}_i$ in the population goes through the mutation operator to form the new individual $\hat{\boldsymbol{X}}_i$, formulated

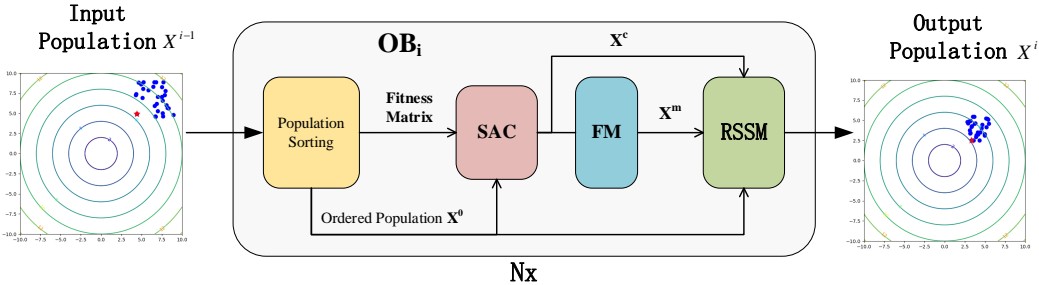

Figure 1: Overall architecture of BOptformer and OB. $Nx$ stands for BOptformer is composed of $Nx$ stacked OBs. These OBs can be set to share weights with each other or not share weights with each other.

as $\hat{X}_i = X_i W_i^m$. $W_i^m$ is the diagonal matrix. In Transformer, each patch embedding carries on directional feature transformation through the FFN module. We take one linear layer as an example: $X = X W^F$, where $W^F$ is the weight of the linear layer, and it is applied to each embedding separately and identically. This equation and the mutation operator have the same formula format, which inspires us to design a learnable mutation module FM based on FFN with $ReLU$ activation function:

$$X^m = FM(X^c) = (ReLU(X W_1^F + b_1)) W_2^F + b_2 \tag{4}$$

where $X^m$ is the population after the mutation of $X^c$. $W_2^F$ and $W_1^F$ represent the weight of the second layer of FFN and the weight of the first layer of FFN, respectively. $b_2$ and $b_1$ represent the bias of the second layer and the first layer of FFN, respectively.

### 3.4 SELECTION MODULE

The residual connection in the transformer can be analogized to the selection operation in EA (Zhang et al., 2021). We combine the residual structure and selection module (SM) (Anonymous, 2023) to design a learnable selection module RSSM. The RSSM generates the offspring population according to the following equation:

$$
\begin{aligned}
\hat{X} &= RSSM(X, X^c, X^m) \\
&= Sort(SM(X, tile(W_1^s) \odot X + tile(W_2^s) \odot X^c + tile(W_3^s) \odot X^m))
\end{aligned}
\tag{5}
$$

where $\hat{X}$ is the fittest population for the next generation; the learnable weights $W_1^s \in \mathbb{R}^{n \times 1}$, $W_2^s \in \mathbb{R}^{n \times 1}$, and $W_3^s \in \mathbb{R}^{n \times 1}$ are the weights for $X$, $X^c$, and $X^m$, respectively. $Sort(X)$ represents that $X$ is sorted in non-descending order of fitness. We use quicksort to sort the population. These three learnable weight matrices realize the weighted summation of residual connections, thereby simulating a learnable selection strategy. Meanwhile, the introduction of residual structure also enhances the model's representation ability, enabling BOptformer to form a deep architecture.

SM updates individuals based on a pairwise comparison between the offspring and input population regarding their fitness. Suppose that $X$ and $X'$ are the input populations of SM. We compare the quality of individuals from $X$ and $X'$ pairwise based on fitness. A binary mask matrix indicating the selected individual can be obtained based on the indicator function $l_{x>0}(x)$, where $l_{x>0}(x) = 1$ if $x > 0$ and $l_{x>0}(x) = 0$ if $x < 0$. SM forms a new population $\hat{X}$ by employing Equation (6).

$$\hat{X} = tile(l_{x>0}(M_{F'} - M_F)) \odot X + tile(1 - l_{x>0}(M_{F'} - M_F)) \odot X' \tag{6}$$

where the *tile* copy function extends the indication vector to a matrix, $M_F(M_{F'})$ denotes the fitness matrix of $X(X')$.

### 3.5 STRUCTURE OF BOPTFORMER

BOptformer comprises basic $t$ BOptformer blocks (OBs), and parameters can be shared among these $t$ OBs or not. The overall architecture of BOptformer and OB is shown in Figure 1. Each OB consists of SAC, FM, and RSSM. $X^0 \in R^{n \times d}$ represents the initial population input into BOptformer, which

needs to be sorted in non-descending order of fitness. In Equation 7, $\boldsymbol{X}^{i-1}$ is fed into $OB_t$ to get $\boldsymbol{X}^i$, where $i \in [1, t]$. BOptformer realizes the mapping from the random initial population to the target population by stacking $t$ OBs.

$$\boldsymbol{X}^i = OB(\boldsymbol{X}^{i-1}); \quad \boldsymbol{X}^c = SAC(\boldsymbol{X}^{i-1}, \boldsymbol{F}); \tag{7}$$
$$\boldsymbol{X}^m = FM(\boldsymbol{X}^c); \quad \boldsymbol{X}^i = RSSM(\boldsymbol{X}^{i-1}, \boldsymbol{X}^c, \boldsymbol{X}^m)$$

### 3.6 TRAINING OF BOPTFORMER

**Training Dataset**  Before introducing the details of the training dataset, fidelity (Kandasamy et al., 2016) is defined as follows: Suppose the differentiable surrogate functions $f_1, f_2, \cdots, f_m$ are the continuous exact approximations of the black-box function $f$. We call these approximations fidelity, which satisfies the following conditions: 1) $f_1, \cdots, f_i, \cdots, f_m$ approximate $f$. $||f - f_i||_\infty \le \zeta_m$, where the fidelity bound $\zeta_1 > \zeta_2 > \cdots \zeta_m$. 2) Estimating approximation $f_i$ is cheaper than estimating $f$. Suppose the query cost at fidelity is $\lambda_i$, and $\lambda_1 < \lambda_2 < \cdots \lambda_m$.

Training data is a crucial factor beyond the objective functions. This paper establishes the training set by constructing a set of differentiable functions related to the optimization objective. This training dataset only contains $(\boldsymbol{X}_0, f_i(\boldsymbol{x}|\omega))$, the initial population and objective function, respectively. The variance of $\omega$ causes the shift in landscapes. The training dataset is designed as follows: 1) Randomly initialize the input population $\boldsymbol{X}_0$; 2) Randomly produce a shifted objective function $f_i(\boldsymbol{x}|\omega)$ by adjusting the parameter $\omega$; 3) Evaluate $\boldsymbol{X}_0$ by $f_i(\boldsymbol{x}|\omega)$; 4) Repeat Steps 1)-3) to generate the corresponding dataset. We show the designed training and testing datasets as follows:

$$F^{train} = \{f_1(\boldsymbol{x}|\omega_{1,i}^{train}), \cdots, f_m(\boldsymbol{x}|\omega_{m,i}^{train})\} \tag{8}$$

where $\omega_{m,i}^{train}$ represents the $i$th different values of $\omega$ in $m$th function $f_m$.

**Loss Function**  BOptformer attempts to search for individuals with high quality based on the available information. The loss function tells how to obtain the parameters of BOptformer to generate individuals closer to the optimal solution by maximizing the difference between the initial population and the output population of BOptformer. The following loss function is employed (Anonymous, 2023),

$$l_i(\boldsymbol{X}^0, f(\boldsymbol{x}|\omega)) = \frac{\frac{1}{|\boldsymbol{X}^0|} \sum\limits_{\boldsymbol{x} \in \boldsymbol{X}^0} f_i(\boldsymbol{x}|\omega) - \frac{1}{|E_\theta(\boldsymbol{X}^0)|} \sum\limits_{\boldsymbol{x} \in E_\theta(\boldsymbol{X}^0)} f_i(\boldsymbol{x}|\omega)}{\left| \frac{1}{|\boldsymbol{X}^0|} \sum\limits_{\boldsymbol{x} \in \boldsymbol{X}^0} f_i(\boldsymbol{x}|\omega) \right|} \tag{9}$$

where $\theta$ denotes parameters of BOptformer ($E$). Equation (9) calculates the average fitness difference between the input and output, further normalized within $[0, 1]$. To encourage BOptformer to explore the fitness landscape, for example, the constructed Bayesian posterior distribution over the global optimum (Cao & Shen, 2020) can be added to Equation (9). Since the derivatives of functions in the training dataset are available, we can obtain the gradient information of Equation (9) for the training process. Also, we can employ REINFORCE (Williams, 1992) to approximate these derivatives.

**Training BOptformer**  We then train BOptformer under a supervised mode. Since the gradient is unnecessary during the test process, BOptformer can solve black-box optimization problems. To prepare BOptformer to learn a balanced performance upon different optimization problems, we design a loss function formulated as follows:

$$l_\Omega = -\frac{1}{K} \sum_{\boldsymbol{X}^0 \in \Omega} l_i(\boldsymbol{X}^0, f_i(\boldsymbol{x}|\omega_i^{train})) \tag{10}$$

We employ Adam (Kingma & Ba, 2014) method with a minibatch $\Omega$ to train BOptformer upon the constructed training dataset.

**Detailed Training Process**  The goal of the training algorithm is to search for parameters $\theta^*$ of the BOptformer. Before training starts, BOptformer is randomly initialized to get initial parameters $\theta$. Then the algorithm will perform the following three steps in a loop until the training termination condition is satisfied: Step 1, randomly initialize a minibatch $\Omega$ comprised of $K$ populations $\boldsymbol{X}^0$;

Step 2, for each $f_i \in F^{train}$, given training data $(\boldsymbol{X}^0, f_i)$, update $\theta$ by minimizing the $l_\Omega$; Step 3, given $\boldsymbol{X}^0$, update $\theta$ by minimizing $-1/m \sum_i l_\Omega$, where $m$ is the number of functions in $F^{train}$. After completing the training process, the algorithm will output $\theta^*$.

## 4 EXPERIMENTS

### 4.1 EXPERIMENTAL SETUP

#### 4.1.1 DATASETS

**Synthetic Functions**    This paper first employs nine commonly used functions to show the effectiveness of the proposed BOptformer. The characteristics of these nine functions are shown in Tables 6 and 7 (Appendix). Here, BOptformer is trained on $F^{train}$ is generated based on functions in Table 6, and the target functions are shown in Table 7 (Appendix). Here, $d = \{10, 100\}$.

**Protein Docking**    We also handle the problem of *Ab initio* protein docking (Cao & Shen, 2020), which optimizes a noisy and costly function in a high-dimensional conformational space. Mathematically, this problem is formulated as optimizing the Gibbs binding free energy $f(\boldsymbol{x})$ for conformation $\boldsymbol{x}$. We calculate the energy function in a CHARMM 19 force field as in (Moal & Bates, 2010) and shift it so that $f(\boldsymbol{x}) = 0$ at the origin of the search space. $f(\boldsymbol{x})$ is differentiable when we parameterize the search space as $\mathbb{R}^{12}$ (Smith & Sternberg, 2002). Here, only 100 interface atoms are considered. The details of this problem can be found in Appendix A.8.

**Planner Mechanic Arm**    We further evaluate the performance of the proposed scheme on the planner mechanic arm problem, which has been widely used to evaluate the performance of the black-box optimization algorithms (Cully et al., 2015; Vassiliades et al., 2018; Vassiliades & Mouret, 2018; Mouret & Maguire, 2020). The optimization goal of this problem is to minimize the distance from the top of the mechanic arm to the target position by optimizing a set of lengths angles. The detailed problem can be found in Appendix A.5. $r$ represents the distance from the target point to the origin of the mechanic arm, as shown in Fig. 4 (Appendix).

Table 1: The compared results on six functions.

| $d$ | $f$ | DE | ES | CMA-ES | L2O-swarm | Dragonfly | BOptformer |
|---|---|---|---|---|---|---|---|
| 10 | F4 | 0.13(0.06) | 0.22(0.30) | 4.2e-4(3.5e-4) | 16.92(2.10) | 1.3e3(1.3e3) | **1.2e-4(5e-5)** |
|  | F5 | 4.99(1.24) | 0.55(0.37) | 0.03(0.01) | 2.97(0.01) | 48.4(9.58) | **8e-3(2e-3)** |
|  | F6 | 210.2(49.7) | 60.02(48.28) | 61.90(96.25) | 26.83(21.48) | 3.8e8(1.4e8) | **8.93(0.03)** |
|  | F7 | 17.83(3.59) | 51.53(8.05) | 45.74(17.02) | 4.88(3.55) | 81.1(24.0) | **0.01(0.03)** |
|  | F8 | 0.21(0.07) | 0.26(0.18) | 7.6e-3(0.01) | 1.02(1.4e-3) | 35.4(22.6) | **1e-5(2e-5)** |
|  | F9 | 1.90(0.32) | 20.56(0.03) | 0.04(0.02) | 9.06(0.67) | 16.2(3.64) | **0.01(3.4e-3)** |
| 100 | F4 | 8.2e3(3.8e2) | 8.9e4(9.5e3) | 7.8e3(1.2e3) | 0.32(0.03) | 11200(3750) | **0.11(0.09)** |
|  | F5 | 28.2(0.61) | 80.2(2.10) | 78.3(9.18) | 0.28(1.3e-3) | 50(0) | **0.14(0.15)** |
|  | F6 | 2.4e8(2.3e7) | 2.5e10(4.5e9) | 3.3e8(8.7e7) | 692(108) | **99(0)** | 129(346) |
|  | F7 | 9410(548) | 8.9e4(1.1e4) | 8050(775) | 85.7(18.6) | 144(13.1) | **24.1(13)** |
|  | F8 | 3.04(0.14) | 23.0(3.00) | 3(0.22) | 0.16(2.4e-4) | 125(11.3) | **0.02(0.03)** |
|  | F9 | 18.9(0.14) | 21.4(0.02) | 21.4(0.04) | 2.49(2.5e-3) | 10.5(0.32) | **0.15(0.05)** |

#### 4.1.2 BASELINES

BOptformer is compared with standard EA baselines, such as DE(DE/rand/1/bin) (Das & Suganthan, 2010), ES($(\mu,\lambda)$-ES), and CMA-ES, where DE and ES are implemented based on geatpy (Jazzbin, 2020), and CMA-ES is implemented by pymoo (Blank & Deb, 2020). L2O-swarm (Cao et al., 2019) is a representative L2O method for black-box optimization. Moreover, Dragonfly (Kandasamy et al., 2020), a representative algorithm for Bayesian optimization, is employed as a reference. We design three BOptformer models, including *3 OBs with WS* (weight sharing), *5 OBs without WS*, and *30 OBs with WS*. The parameters of these methods are shown in Appendix A.4.

## 4.2 Results

**Synthetic Functions** The results on six functions are provided in Table 1. BOptformer outperforms three EA baselines, Dragonfly, and L2O-swarm in all cases, but loses once to Dragonfly in F6 with $d = 100$. These cases also show the excellent generalization ability of BOptformer on more tasks unseen during the training stage. We think the transferability of BOptformer is proportional to the fitness landscape similarity between the training set and the problem. Although new problem attributes are not available in the training set, BOptformer can still perform better. However, this conclusion only holds when the similarity between the problem and training dataset is high. We plot the convergence curves of BOptformer (10 OBs with WS), ES, DE, and CMA-ES on F7 (see Appendix A.7, Figure 5). BOptformer converges quickly and can obtain better solutions. BOptformer can only iterate ten times to get the best solution relative to EA baselines. ES and DE converge around 100 generations, and CMA-ES shows a slow convergence rate.

**Protein Docking** We also test the performance of BOptformer on the problem of protein docking. The experimental results is shown in 2. The performance of BOptformer exceeds that of L2O-swarm.

Table 2: The results on the problem of protein docking.

| Methods | 1ATN_7 | 2JEL_1 | 7CEI_1 |
|---|---|---|---|
| L2O-swarm | 2090(25.08) | 2765(24.80) | 1689(23.64) |
| *30OBs with WS* | -6.03e3(145) | -6.03e3(127) | -5.97e3(125) |
| *5OBs without WS* | -6.01e3(146) | -6.02e3(125) | -5.98e3(131) |

During training, L2O-swarm does not converge. At the same time, we find that better solutions exist in the initial population than those found by L2O-swarm. However, during testing, L2O-swarm lost these good solutions.

**Planner Mechanic Arm** The detailed experimental results are given in Tables 3. BOptformer selects *5 OBs without WS* as the example, which evolves only five generations. *Untrained* represents the untrained BOptformer. DE, ES, and CMA-ES are tested when the maximum generations is set to 100. EA baselines have 100/5 times as many function evaluations as BOptformer. However, even in this unfair situation, BOptformer achieves the best results. We have observed that BOptformer can achieve better results with deeper architectures. However, it is currently difficult for us to train deep BOptformer. Moreover, as far as we know, the use of ES to optimize deep models has been studied a lot (Vicol et al., 2021), which will be an essential research prospect in the future.

Table 3: The results of planar mechanical arm. Simple Case (SC): searching for different angles with the fixed lengths. Complex Case (CC): searching for different angles and lengths.

| | $r$ | DE | ES | CMA-ES | L2O-Swarm | BOptformer | *Untrained* |
|---|---|---|---|---|---|---|---|
| | 100 | 1.20(0.64) | 10.6(5.58) | 1.36(0.35) | 40.4(3.89) | **0.30(0.18)** | 243(238) |
| SC | 300 | 1.38(0.71) | 44.9(43.3) | 1.38(0.41) | 69.5(3.77) | **0.48(0.37)** | 1210(820) |
| | 1000 | 93.8(137) | 183(239) | 43.7(110) | 176(7.20) | **26.6(57.4)** | 5070(2770) |
| | 100 | 0.81(0.47) | 8.95(6.42) | 0.76(0.20) | 31.9(1.78) | **0.06(0.05)** | 243(238) |
| CC | 300 | 6.15(12.2) | 47.8(56.0) | 0.87(0.37) | 89.1(1.96) | **0.50(0.79)** | 1210(820) |
| | 1000 | 232(233) | 251(258) | 88.4(158) | 262(2.99) | **25.0(55.8)** | 5070(2770) |

## 4.3 Parameter Analysis

We consider the performance of different BOptformer architectures. Here, $d = 10$. The experimental results are shown in Table 4. We find that they were sorted from good to worst by their performance, and the result is *30 OBs with WS>5 OBs without WS>3 OBs with WS*. Deep architectures have better representation capabilities and also lead to better per-

Table 4: The performance of different BOptformer structures.

| $f$ | *Untrained* | *5 OBs without WS* | *30 OBs with WS* | *3 OBs with WS* |
|---|---|---|---|---|
| F4 | 0.28(0.09) | 0.08(0.03) | **1.2e-4(5e-5)** | 8.16(3.44) |
| F5 | 0.37(0.05) | 0.15(0.03) | **0.008(0.002)** | 1.47(0.40) |
| F6 | 45.8(16.9) | 15.43(2.34) | **8.93(0.03)** | 1891(1396) |
| F7 | 1.08(0.72) | 4.43(1.82) | **0.01(0.03)** | 35.72(8.52) |
| F8 | 0.69(0.09) | 0.06(0.03) | **1e-5(2e-5)** | 0.82(0.10) |
| F9 | 0.85(0.20) | 0.29(0.07) | **0.01(0.003)** | 3.28(1.00) |

formance. However, it is challenging to train non-weight sharing BOptformers with more layers due to the difficulty of training deep architectures. *Untrained* represents that the parameters of *5 OBs without WS* are randomly initialized. The results show that *5 OBs without WS* outperforms *Untrained*, which demonstrates the effectiveness of the designed training process.

Table 5: The results of ablation study. $d = 10$.

| $f$ | Not SAC | Not FM | Not RC | Not RSSM | BOptformer |
|---|---|---|---|---|---|
| F4 | 52.7(15.0) | 23.3(6.68) | 50.5(6.81) | 271(346) | 3.03(5.82) |
| F5 | 5.08(0.98) | 2.94(0.41) | 3.75(0.09) | 3.51(1.49) | 1.02(0.20) |
| F6 | 1.25e5(9.26e4) | 1.10e4(1.44e4) | 5.22e4(1.28e4) | 6.99e6(1.21e8) | 4.03e3(6.94e4) |
| F7 | 47.2(5.11) | 19.7(4.27) | 63.4(8.60) | 40.6(8.10) | 44.8(8.34) |
| F8 | 1.18(0.04) | 1.19(0.07) | 0.87(0.07) | 3.28(1.14) | 0.60(0.17) |
| F9 | 4.49(1.09) | 3.42(0.78) | 8.36(0.43) | 3.56(0.72) | 3.03(0.70) |

We also find an interesting phenomenon: *5 OBs without WS* outperforms *3 OBs with WS* in all cases. Our untrained deep architecture, *5 OBs without WS*, can achieve good results on simple cases, which shows that BOptformer retains the advantages of Transformer architecture and has strong generalization ability. We use the untrained *5 OBs with WS* to test on the complex plannar mechanic arm problem and find that it performs poorly.

We train BOptformer on the F1-F3 function set with different learning rates ($lr$) and then test them on the F4-F9 function set. The experimental results are shown in Table 9 (Appendix A.6). For *5 OBs without WS*, setting $lr = 0.01$ achieves the relatively best performance. Using $lr = 0.0001$ would be a good choice for *30 OBs with WS* and *3 OBs with WS*.

## 4.4 Ablation Study

This section considers the performance impact of different parts in BOptformer. We take BOptformer with 3 OBs and weight sharing as an example, which is trained on F1-F3 and tested on F4-F9. We remove SAC, FM, RSSM, and RC in BOptformer, respectively, and denote them as *Not SAC, Not FM*, *Not RSSM*, and *Not RC*. The experimental results are shown in Table 5. When their results were sorted from good to worst, the rank is BOptformer > *Not FM* > *Not RC* ≈ *Not SAC* ≈ *Not RSSM*. The role of FM is slightly weaker than that of the other three modules. Taken as a whole, the parts of SAC, RSSM, and RC are of equal importance. The absence of these core components can seriously affect the performance of BOptformer. At the same time, it also shows the effectiveness of the proposed four modules. The removal of any one of the modules in the crossover, mutation, and selection of EAs will degrade the performance of EAs. This shows that BOptformer implements a learnable EA framework that does not require human-designed parameters.

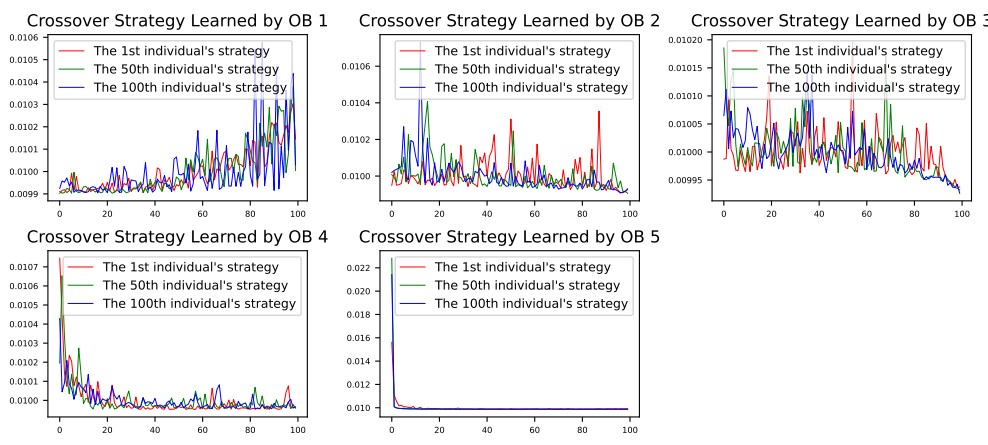

Figure 2: Crossover Strategy learned by BOptformer.

## 4.5 Visualization analysis

The tested model is *5 OBs with WS* trained on F1-F3 with $d = 100$. The population size is 100.

**Visual Analysis of SAC** The crossover strategies learned by the five SAC are shown in Fig. 2. For the presentation, we select individuals with fitness rankings 1st, 50th, and 100th. The horizontal axis

represents the fitness ranking of individuals, and the vertical axis represents the attention (weight when performing crossover) on these individuals. OB1 tends to crossover with lower-ranked individuals, showing a preference for exploration. From OB1 to OB5, the bias of SAC gradually changes from exploration to exploitation.

**Visual Analysis of FM** We test *5 OBs with WS* on F4 with $d = 2$. The mutation strategies learned by the five OBs are shown in Fig. 3. Input and output represent the input and output populations of the FM module, respectively. OB1 tends to explore a broad solution space, and the next 4 OBs gradually shift from searching the vast space to searching the space near the input population. The strategies learned by FM and SAC modules show a common feature: the preference for generating solutions gradually shifts from exploration to exploitation as the population converges.

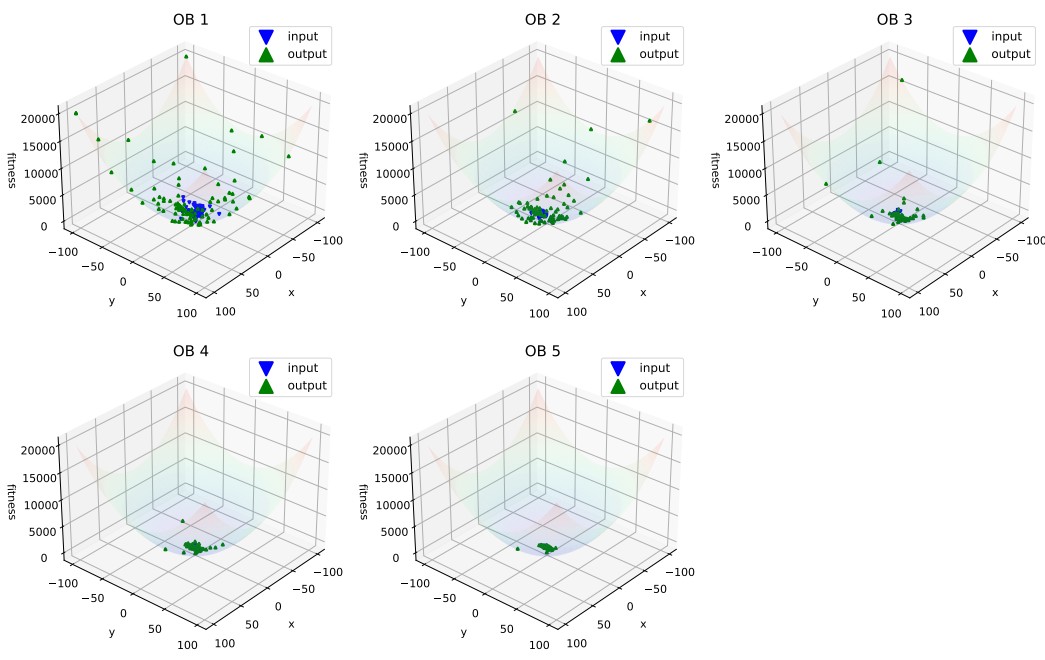

Figure 3: Mutation strategy learned by BOptformer.

## 5 CONCLUSIONS

We successfully designed the Transformer-based L2O framework for black-box optimization, which does not need hand-designed operators. The better performance than that of EA baselines, Bayesian optimization, and the L2O method demonstrates the effectiveness of BOptformer. Moreover, BOptformer can be well adapted to unseen black-box optimization. Meanwhile, we experimentally demonstrate that the proposed three modules have positive effects. BOptformer still has room for improvement.

1) Our scheme is not limited to black-box optimization. Similar to the LSTM architecture, our scheme can directly optimize differentiable functions. However, the architecture of BOptformer does not directly involve the gradient information of the optimization target, which makes BOptformer inferior to existing L2O schemes. In future work, we will design a new module that embeds the gradient information of the optimization target;

2) In the loss function, we did not effectively consider the diversity of the population, and the population can be regularized in the future;

3) The training set seriously affects the performance of BOptformer. If the similarity between the training set and the optimization objective is low, it will cause the performance of BOptformer to degrade drastically. Building the dataset as relevant to the target as possible is essential.

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

# A APPENDIX

## A.1 VISION TRANSFORMER

We mainly introduce the core part of Vision Transformer, such as the multi-head self-attention layer (MSA), feed-forward network (FFN), layer normalization (LN), and residual connection (RC).

**MSA** MSA fuses several SA operations to handle the queries ($Q$), keys ($K$), and values ($V$) that jointly attend to information from different representation subspaces. MSA is formulated as follows: $MultiHead(Q, K, V) = Concat(H_1, H_2, \cdots H_h)W^O$. where $Concat$ means concatenation operation. The head feature $H_i$ can be formulated as:

$$H_i = SA(QW_i^Q, KW_i^K, VW_i^V)$$
$$= Softmax\left(QW_i^Q(KW_i^K)^T/sqrt(d_k)\right)VW_i^V = AVW_i^V$$

where $W_i^Q \in \mathbb{R}^{d_m \times d_q}$, $W_i^K \in \mathbb{R}^{d_m \times d_k}$, and $W_i^V \in \mathbb{R}^{d_m \times d_v}$ are parameter matrices for queries, keys, and values, respectively; $W^O \in \mathbb{R}^{hd_v \times d_m}$ maps each head feature $H_i$ to the output. Moreover, $d_m$ is the input dimension, while $d_q, d_k$, and $d_v$ are hidden dimensions of the corresponding projection subspace; $h$ is the head number. $A \in \mathbb{R}^{l \times l}$ is the attention matrix of $h$th head, $l$ is the sequence length.

**FFN** FFN employs two cascaded linear transformations with a ReLU activation to handle $X$, which is shown as: $FFN(X) = max(0, XW_1 + b_1)W_2 + b_2$, where $W_1$ and $W_2$ are weights of two linear layers, and $b_1$ and $b_2$ are corresponding biases.

**LN** LN is applied before each layer of MSA and FFN, and the output of LN is calculated by $X + [MSA|FFN](LN(X))$.

## A.2 PRELIMINARY EAS

The crossover, mutation, and selection operators form the basic framework of EAs. EA starts with a randomly generated initial population. Then, genetic operations such as crossover and mutation will be carried out. After the fitness evaluation of all individuals in the population, a selection operation is performed to identify fitter individuals to undergo reproduction to generate offspring. Such an evolutionary process will be repeated until specific predefined stopping criteria are satisfied.

**Crossover** The crossover operator generates a new individual $\hat{X}_i$ by Equation (11), and $cr$ is the probability of the crossover operator.

$$\hat{X}_{i,k}^c = \begin{cases} X_{j,k} & rand(0,1) < cr \\ X_{i,k} & otherwise \end{cases} \tag{11}$$

where $k \in [1, \cdots, d]$. This operator is commonly conducted on $n$ individuals. After an expression expansion, we re-formulate Equation (11) as $\sum_{i=1}^{n} X_i W_i^c$ (Zhang et al., 2021). $W_i^c$ is the diagonal matrix. If $W_i^c$ is full of zeros, the $i$th individual has no contribution.

**Mutation** The mutation operator brings random changes into the population. Specifically, an individual $X_i$ in the population goes through the mutation operator to form the new individual $\hat{X}_i$, formulated as follows:

$$\hat{X}_{i,k}^m = \begin{cases} rand(l_k, u_k) & rand(0,1) < mr \\ \hat{X}_{i,k}^c & otherwise \end{cases} \tag{12}$$

where $mr$ is the probability of mutation operator and $k \in [1, \cdots, d]$. Similarly, Equation (12) can be re-formulated as $X_i W_i^m$, where $W_i^m$ is the diagonal matrix.

**Selection** We introduce the binary tournament mating selection operator in Equation (13). The selection operator survives individuals of higher quality for the next generation until the number of individuals is chosen.

$$p_i = \begin{cases} 1 & f(\boldsymbol{X}_i) < f(\boldsymbol{X}_k) \\ 0 & f(\boldsymbol{X}_i) > f(\boldsymbol{X}_k) \end{cases}, \quad (\boldsymbol{X}_i, \boldsymbol{X}_k) \in \boldsymbol{X}, \tag{13}$$

where $p_i$ reflects the probability that $\boldsymbol{X}_i$ is selected for the next generation, and $(\boldsymbol{X}_i, \boldsymbol{X}_k)$ in Equation (13) are randomly selected from the population $\boldsymbol{X} \cup \hat{\boldsymbol{X}}^m$.

### A.3 SYNTHETIC FUNCTIONS

Table 6: Training functions.

| ID | Functions | Range |
|----|-----------|-------|
| F1 | $\sum_i \lvert w_i sin(x_i - b_i) \rvert$ | $x \in [-10, 10], b \in [-10, 10]$ |
| F2 | $\sum_i \lvert x_i - b_i \rvert$ | $x \in [-10, 10], b \in [-10, 10]$ |
| F3 | $\sum_i \lvert (x_i - b_i) + (x_{i+1} - b_{i+1}) \rvert + \sum_i \lvert x_i - b_i \rvert$ | $x \in [-10, 10], b \in [-10, 10]$ |

Table 7: Testing Functions.

| ID | Functions | Range |
|----|-----------|-------|
| F4(Sphere) | $\sum_i z_i^2, z_i = x_i - b_i$ | $x \in [-100, 100], b \in [-50, 50]$ |
| F5 | $\max\{\lvert z_i \rvert, 1 \le i \le D\}, z_i = x_i - b_i$ | $x \in [-100, 100], b \in [-50, 50]$ |
| F6(Rosenbrock) | $\sum_{i=1}^{D-1} (100(z_i^2 - z_{i+1})^2 + (z_i - 1)^2), z_i = x_i - b_i$ | $x \in [-100, 100], b \in [-50, 50]$ |
| F7(Rastrigin) | $\sum_{i=1}^{D} (z_i^2 - 10\cos(2\pi z_i) + 10), z_i = x_i - b_i$ | $x \in [-5, 5], b \in [-2.5, 2.5]$ |
| F8(Griewank) | $\sum_{i=1}^{D} \frac{z_i^2}{4000} - \prod_{i=1}^{D} \cos(\frac{z_i}{\sqrt{i}}) + 1, z_i = x_i - b_i$ | $x \in [-600, 600], b \in [-300, 300]$ |
| F9(Ackley) | $-20\exp(-0.2\sqrt{\frac{1}{D}\sum_{i=1}^{D} z_i^2}) - \exp(\frac{1}{D}\sum_{i=1}^{D} \cos(2\pi z_i)) + 20 + \exp(1), z_i = x_i - b_i$ | $x \in [-32, 32], b \in [-16, 16]$ |

### A.4 PARAMETERS

**BOptformer**. For example, *30 OBs with WS* contains 30 OBs, and each OB consists of 1 SAC, 1 FM, and 1 RSSM. In *30 OBs with WS*, these 30 OBs share parameters. *5 OBs without WS* has 5 OBs, and no parameters are shared among them. During the training process, BOptformer is iterated for 1000 epochs. The initial learning rate ($lr$) was set to 0.01 and $lr = lr \times 0.9$ each 100 cycles. The 2-norm of the gradient is clipped so that it is not larger than 10. The bias of the function is regenerated each epoch, and a new batch of random initial populations is generated.

**Baselines**. The number of generations of the reference algorithms is set to 100. The population size of ES, DE, and CMA-ES is set to 100. For all cases, we choose the optimal hyperparameters. To ensure validity, all experimental results are averaged over 10 runs. All experiments were performed on a Ubuntu20.04 PC with Intel(R) Core I7 (TM) I3-8100 CPU at 3.60GHz and NVIDIA GeForce GTX 1060.

### A.5 PLANNER MECHANIC ARM PROBLEM

The optimization goal of this problem is to search for a set of lengths $L = (L_1, L_2, \cdots, L_n)$ and a set of angles $\alpha = (\alpha_1, \alpha_2, \cdots, \alpha_n)$ so that the distance $f(L, \alpha, p)$ from the top of the mechanic arm to the target position $p$ is the smallest, where $n$ represents the number of segments of the mechanic

arm, and $L_i \in (l_i, u_i)$ and $\alpha_i \in (-\Pi, \Pi)$ represent the length and angle of the $i$th mechanic arm, respectively. Typically, $d$ is calculated as follows:

$$f(L, \alpha, p) = \sqrt{\left(\sum_{i=1}^{n} \cos(\alpha_i) L_i - p_x\right)^2 + \left(\sum_{i=1}^{n} \sin(\alpha_i) L_i - p_y\right)^2} \tag{14}$$

where $p_x$ and $p_y$ represent the x-coordinate and y-coordinate of the target point, respectively.

Here, $n = 100$, $l_i = 0$ and $u_i = 10$. We design two groups of experiments. *1) Simple case*. We fixed the length of each mechanic arm as $l_i = 10$ and only searched for the optimal $\alpha$. *2) Complex case*. We need to search for $L$ and $\alpha$ simultaneously. We randomly selected 600 target points within the range of $r \leq 1000$ to form a set $S$, where $r$ represents the distance from the target point to the origin of the mechanic arm, as shown in Fig. 4. During the training process of BOptformer, a sample point set $s$ is re-extracted from $S$ for training every $T$ training cycle. In the testing process, we extracted 128 target points ($S^{test}$) in the range of $r \leq 100$, $r \leq 300$, and $r \leq 1000$, respectively, for testing. The purpose of testing in three different regions is to explore the generalization performance of BOptformer further. We evaluate the generalization ability of the algorithm by $\left(\sum_{s}^{S^{test}} f(L, \alpha, s)\right) / |S^{test}|$.

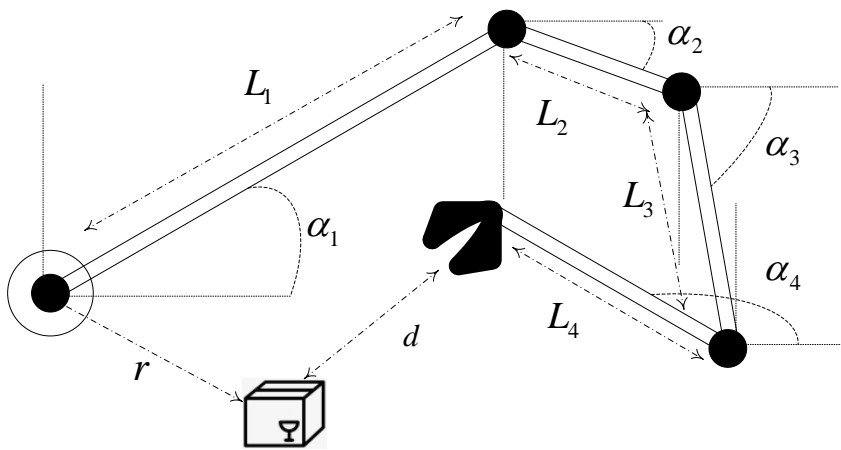

Figure 4: Planar Mechanical Arm.

DE, ES, and CMA-ES are tested when the maximum generations $Maxgen$ is set to 10, 50, and 100, respectively. We find that BOptformer outperforms all baselines.

### A.6 EFFECT OF LEARNING RATE ON BOPTFORMER

We train BOptformer on the F1-F3 function set with different learning rates, and then test it on the F4-F9 function set. The experimental results are shown in Table 9. *5 OBs without WS* and *30 OBs with WS* perform poorly when the learning rate is 0.1, which may be because the learning rate is too large, which affects the convergence of BOptformer during the training process. For *5 OBs without WS*, setting the learning rate to 0.01 achieves relatively best performance. Using a learning rate of 0.0001 would be a good choice for *30 OBs with WS* and *3 OBs with WS*. However, our experiments are coarse-grained. The learning rate has a greater impact on BOptformer. Then using Auto-ML to search for the optimal hyperparameter combination of the model is expected to achieve better performance.

### A.7 CONVERGENCE OF BOPTFORMER

We plot the convergence curves of *30 OBs with WS*, ES, DE, and CMA-ES on F7. BOptformer converges quickly and can obtain better solutions. BOptformer can only iterate ten times to get the best solution relative to EA baselines. ES and DE converged around 100 generations, and CMA-ES showed a slow convergence rate.

Table 8: The results of planar mechanical arm on searching for different angles with the fixed lengths.

| | Maxgen=10 | | | | |
|---|---|---|---|---|---|
| $r$ | DE | ES | CMA-ES | L2O-Swarm | BOptformer |
| 100 | 2.96(1.63) | 11.2(4.70) | 236(46.8) | 40.4(3.89) | **0.30(0.18)** |
| 300 | 11.3(14.7) | 45.3(43.3) | 243(125) | 69.5(3.77) | **0.48(0.37)** |
| 1000 | 227(226) | 257(246) | 397(321) | 176(7.20) | **26.6(57.4)** |
| | Maxgen=50 | | | | |
| $r$ | DE | ES | CMA-ES | L2O-Swarm | BOptformer |
| 100 | 1.28(0.60) | 10.7(5.91) | 2.42(0.65) | 40.4(3.89) | **0.30(0.18)** |
| 300 | 1.54(0.89) | 42.0(41.0) | 4.06(6.54) | 69.5(3.77) | **0.48(0.37)** |
| 1000 | 110(152) | 193(235) | 103(182) | 176(7.20) | **26.6(57.4)** |
| | Maxgen=100 | | | | |
| $r$ | DE | ES | CMA-ES | L2O-Swarm | BOptformer |
| 100 | 1.20(0.64) | 10.6(5.58) | 1.36(0.35) | 40.4(3.89) | **0.30(0.18)** |
| 300 | 1.38(0.71) | 44.9(43.3) | 1.38(0.41) | 69.5(3.77) | **0.48(0.37)** |
| 1000 | 93.8(137) | 183(239) | 43.7(110) | 176(7.20) | **26.6(57.4)** |

Table 9: Ablation study on learning rate.

| lr | F4 | F5 | F6 | F7 | F8 | F9 |
|---|---|---|---|---|---|---|
| *5 OBs without WS* | | | | | | |
| 0.1 | 0.93(7.42) | 0.31(0.49) | 2.04e7(2.03e8) | 15.3(6.4) | 0.36(0.16) | 0.61(0.18) |
| 0.01 | **0.01(0.003)** | **0.05(0.02)** | **9.57(0.22)** | **1.62(0.60)** | **0.03(0.01)** | **0.06(0.03)** |
| 0.001 | 0.88(3.41) | 0.36(0.12) | 226(1750) | 6.18(2.66) | 0.56(0.16) | 1.36(0.36) |
| 0.0001 | 0.06(0.03) | 0.13(0.03) | 13.6(2.11) | 0.83(0.50) | 0.17(0.04) | 0.28(0.10) |
| *30 OBs with WS* | | | | | | |
| 0.1 | 1.64(1.19) | 0.85(1.59) | 493(3110) | 28.4(4.35) | 0.47(0.11) | 2.82(0.5) |
| 0.01 | 0.05(0.30) | 0.09(0.03) | 39.2(240) | 1.05(1.40) | 0.01(0.06) | 0.28(0.07) |
| 0.001 | **1.01e-3(0.001)** | 0.02(0.01) | 9.03(0.18) | 0.03(0.02) | **0.003(0.001)** | 0.03(0.01) |
| 0.0001 | 1.50e-3(0.001) | **0.016(0.004)** | **9.01(0.13)** | **0.02(0.02)** | 0.006(0.002) | **0.02(0.0.01)** |
| *3 OBs with weights sharing* | | | | | | |
| 0.1 | 3.04(5.55) | 0.98(0.4) | 1150(7930) | 43.3(8.87) | **0.64(0.12)** | 2.43(0.98) |
| 0.01 | 29.9(47.7) | 2.68(1.30) | 6.24e4(4.87e5) | 40.4(8.52) | 1.05(0.06) | 4.45(0.85) |
| 0.001 | 1.82(1.20) | 0.76(0.32) | 654(4780) | 7.00(7.11) | 0.79(0.13) | 1.91(0.53) |
| 0.0001 | **0.39(0.21)** | **0.33(0.07)** | **46.8(79.4)** | **2.22(2.41)** | 0.66(0.09) | **0.59(0.19)** |

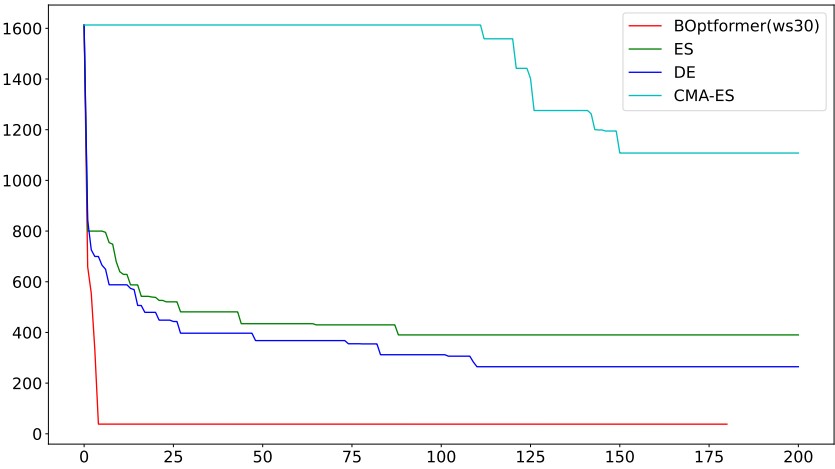

Figure 5: Convergence curves of Boptformer and EA baselines. The figure shows the convergence curve of these algorithms on F7 in Table 7.

## A.8   The Details of Protein Docking

**Protein Docking**    We also handle the problem of *Ab initio* protein docking (Cao & Shen, 2020), which optimizes a noisy and costly function in a high-dimensional conformational space. Mathematically, this problem is formulated as optimizing the Gibbs binding free energy $f(\boldsymbol{x})$ for conformation $\boldsymbol{x}$. We calculate the energy function in a CHARMM 19 force field as in (Moal & Bates, 2010) and shift it so that $f(\boldsymbol{x}) = 0$ at the origin of the search space. $f(\boldsymbol{x})$ is differentiable when we parameterize the search space as $\mathbb{R}^{12}$ (Smith & Sternberg, 2002). Here, only 100 interface atoms are considered.

*Training dataset*.   25 protein-protein complexes (see Appendix A.8) from the protein docking benchmark set 4.0 (Hwang et al., 2010), each of which has 5 starting points (top-5 models from ZDOCK (Pierce et al., 2014)).

*Testing dataset*.   Three complexes (with one starting model each) of different levels of docking difficulty are selected, including 1ATN_7, 2JEL_1, and 7CEI_1.

**25 Protein-protein Complexes**    The training dataset contains 25 protein-protein complexes from the protein docking benchmark set 4.0 (Hwang et al., 2010). The detailed information is shown as follows: 1ATN, 1AVX, 1AY7, 1BJ1, 1BVN, 1CGI, 1DFJ, 1EAW, 1EWY, 1EZU, 1GRN, 1IBR, 1IJK, 1IQD, 1JPS, 1KXQ, 1M10, 1MAH, 1N8O, 1PPE, 1R0R, 1XQS, 2B42, 2C0L, and 2HRK.

