# OpenReview forum: "Optformer: Beyond Transformer for Black-box Optimization"
_ICLR.cc/2023/Conference — Submitted to ICLR 2023_

### Official Review · Reviewer_xa7p · 2022-10-18

**Confidence:** 3
**Correctness:** 3
**Technical Novelty And Significance:** 2
**Empirical Novelty And Significance:** 2
**Recommendation:** 5

**Clarity, Quality, Novelty And Reproducibility:**

## Quality + Clarity:
Please see my above first point on "Weaknesses". Overall the paper can be made much more clear and polished following my suggestions for improvement.

## Originality:
Moderate originality. As referenced already in this paper, there are multiple previous works which e.g. use LSTMs to perform blackbox optimization. One core contribution the current paper makes from previous works, is the interpretability of the Transformer block as an actual evolutionary population update. However, the paper does not experimentally ablate this contribution very well, which is unfortunate.

**Strength And Weaknesses:**

# Strengths
* The overall approach (i.e. a Transformer block has an analogy to an evolutionary algorithm population update) is interesting. If the weaknesses below are resolved, this could be a solid contribution to the field of evolutionary algorithms.
* The baselines used in the experiments (e.g. ES, CMA-ES, Dragonfly) are quite comprehensive. This at least currently establishes that the method has some promise in end-to-end performance.


# Weaknesses
* Experimental results do not present how the intermediate layers work, but only present the end result. This makes it hard to know if the method is doing what it's supposed to, especially given the large claims about the model's analogies to evolutionary algorithms. For example, some lingering questions exist, and can be resolved with more ablations:
    * What kinds of population updates / cross-over techniques is each block learning? Are they similar or different from baselines?
    * "Perhaps using ES is a good training strategy": Is the untrained block performing some sort of local or global mutation?
* Currently, the paper only presents work on continuous search spaces. However, one big advantage of using regular evolutionary algorithms are their flexibility in dealing with combinatorial search spaces. Could the method be modified to approach combinatorial / categorical spaces?
* Large amounts of unnecessary notation make it very difficult to comprehend what is going on. In general, it's not hard to convey my summary written above with fewer notation in general, and this could significantly improve the paper's quality and presentation. For example:
    * Section 3.2 can be first introduced by stating the intent (i.e., showing that crossover / mutations can be represented as matrix multiplications).
    * Section 3.3 can be moved to the Appendix or deleted altogether since they simply describe the original Transformer module.
    * It would also make a huge difference in readability, if e.g. a diagram was made to represent Sections 4.1-4.3 (e.g. see Figure 1 in the original Transformer paper [1]).
    * Too many abbreviations (e.g. "nws5") in the experimental section in Section 5. These could simply be written explicitly, e.g. "Trained"/"Untrained/"5 layers".
* (Minor): It seems that a previous paper [2] has already claimed the "OptFormer" name. I would suggest changing "OptFormer" in this paper to a different name.



[1] Attention is all you need. https://arxiv.org/abs/1706.03762

[2] Towards Learning Universal Hyperparameter Optimizers with Transformers. https://arxiv.org/abs/2205.13320

**Summary Of The Paper:**

# Summary
At a high level, the paper in order can be summarized as:

1. Operations (e.g. crossover) in evolutionary algorithms over continuous search spaces, can be seen as specific matrix multiplications over a population of suggestions $X$.
2. This implies a form of learnability over these matrix multiplications, and a modified variant of the Transformer block can be interpreted as a large, learnable population mutator, mapping an initial population $X_{t}$ to an intermediate population $X_{t+1}$. These blocks can be stacked together to obtain population chains $X_{0} \rightarrow X_{1} \rightarrow X_{2} ...$. Note that then the output of this "modified Transformer" will be the final population.
3. The model is trained directly with a basic normalized performance loss, over **differentiable** functions.
4. Experiments are conducted to check the end-to-end performance of the method, which seems to outperform standard evolutionary algorithm baselines. Ablations are also conducted to show:
    * Deeper models perform better.
    * A deep randomly initialized model "nws5-r" outperforms a trained shallow model "ws3"

**Summary Of The Review:**

Overall, while the approach and overall theme are solid, the execution of the paper is still currently flawed and needs much polishing (both in the writing/presentation, in addition to needing more experimental studies). Thus I currently recommend rejection.

---

> ### Author Response · Authors · 2022-11-18
> **Response to Reviewer xa7p (Part 2)**
>
> **Q3:** Currently, the paper only presents work on continuous search spaces. However, one big advantage of using regular evolutionary algorithms are their flexibility in dealing with combinatorial search spaces. Could the method be modified to approach combinatorial / categorical spaces?
>
> **Answer 3:** The question you raised is the future research direction of BOptformer.
>
> **Solution 1:** Modify the module of BOptformer so that it can be adapted to solve in discrete space. Modify the SAC module to ensure that information can be exchanged between individuals at the same location but not between different locations. For the FM module, the value range of each dimension of the solution is considered to complete the personalized mutation. The construction of the RSSM, loss function, and training set does not need to change.
>
> **Solution 2:** Convert the discrete problem into a continuous problem. For example, for the traveling salesman problem (TSP), we can use GNN to extract the embedded representation of the graph and then use BOptformer to solve the TSP problem on the embedded representation of the graph.
>
> **Q4:** Large amounts of unnecessary notation make it very difficult to comprehend what is going on. In general, it's not hard to convey my summary written above with fewer notation in general, and this could significantly improve the paper's quality and presentation.
>
> **Answer 4:** Thank you for your comments. We have made corresponding changes. Please review our revised version for details. The blue fonts indicate the revised content. The modifications can be summarized as follows:
> 1) We renamed Optformer to BOptformer. The work in [1] is interesting. In the revised version, we also cite this work.
> 2) We remove the extensive introduction to evolutionary algorithms and VIT in Section 3. We focus on our proposed model and analyze experimental results.
> 3) Section 3.2 and Section 3.3 are moved to the Appendix.
> 5) We have redrawn the model architecture diagram (see Figure 1 in the revised version).
> 6) We add experimental results on the complex planner mechanic arm problem in Section 4.2 to further verify the effectiveness of our scheme.
> 7) We add a parametric analysis of the learning rate in Section 4.3 and show the results for different learning rates in the Appendix.
> 8) We visualize the learned crossover and mutation strategies in Section 4.5.
> 9) The abbreviations in the experimental section have been written explicitly (see Section 4).
>
> [1] Towards Learning Universal Hyperparameter Optimizers with Transformers. https://arxiv.org/abs/2205.13320
>
> **Q4:** As referenced already in this paper, there are multiple previous works which e.g. use LSTMs to perform blackbox optimization. One core contribution the current paper makes from previous works, is the interpretability of the Transformer block as an actual evolutionary population update. However, the paper does not experimentally ablate this contribution very well, which is unfortunate.
>
> **Answer 5:** Thank you for your valuable comments. We have compared our proposal with the advanced L2O-Swarm (LSTM-based black-box optimization method) in the paper. Regarding synthetic Functions, protein docking, and planar mechanic arm problems, our scheme is superior to L2O-Swarm. Our newly added results on the planar mechanic arm problem further validate the effectiveness of our scheme (see Section 4.2, Table 3). The experimental results are as follows:
>
> |  Case | $r$ | DE | ES | CMA-ES | L2O-Swarm | nws5 | Untrained|
> | - | - | - | - | - | - | - | -|
> |SC | 100 | 1.20(0.64) | 10.6(5.58) | 1.36(0.35) | 40.4(3.89) | **0.30(0.18)** | 243(238)|
> |SC | 300 | 1.38(0.71) | 44.9(43.3) | 1.38(0.41) | 69.5(3.77) | **0.48(0.37)** | 1210(820)|
> |SC | 1000 | 93.8(137) | 183(239) | 43.7(110) | 176(7.20) | **26.6(57.4)** | 5070(2770)|
> |CC | 100 | 0.81(0.47) | 8.95(6.42) | 0.76(0.20) | 31.9(1.78) | **0.06(0.05)** | 243(238)|
> | CC | 300 | 6.15(12.2) | 47.8(56.0) | 0.87(0.37) | 89.1(1.96) | **0.50(0.79)** | 1210(820) |
> |CC | 1000 | 232(233) | 251(258) | 88.4(158) | 262(2.99) | **25.0(55.8)** | 5070(2770)|

---

> ### Author Response · Authors · 2022-11-18
> **Response to Reviewer xa7p (Part 1)**
>
> Thanks for your positive review of our article! Your comments make our article a qualitative improvement. We have also uploaded the revised version, and we kindly ask you to re-evaluate our paper.
>
> **Q1:** Experimental results do not present how the intermediate layers work, but only present the end result. This makes it hard to know if the method is doing what it's supposed to, especially given the large claims about the model's analogies to evolutionary algorithms. For example, some lingering questions exist, and can be resolved with more ablations.
> What kinds of population updates / cross-over techniques is each block learning? Are they similar or different from baselines?
>
> **Answer 1**: Based on your suggestion, we visualize and analyze the policies learned by the OB layer of BOptformer (nws5) (see Section 4.5, Figures 2 and 3).
>
> Figure 2 shows the mutation strategy learned by the 5 OBs (actually FM) of BOptformer (nws5). The mutation strategy of OB1 tends to explore a broad solution space, and the mutation strategies of the next few OBs gradually shift from searching the vast space to searching the nearby space of the input population of OB.
>
> Figure 3 shows the crossover strategies learned by the 5 OBs (actually SACs) of BOptformer (nws5). For the presentation, we select individuals with fitness ranks 1st, 50th, and 100th. The crossover strategy exhibited by OB1 tends to cross with lower-ranked individuals (individuals with poor fitness), showing a preference for exploration. From OB1 to OB5, the bias of the crossover strategy is gradually transformed from exploration to utilization. The changes offered by the crossover strategy are similar to the changes shown by the mutation strategy learned by OB; that is, the exploration is emphasized first, and then the exploitation is biased.
>
> However, traditional EA does not consider these. As the iteration continues, exploration and exploitation are considered equally. This is one of the reasons why BOptformer is better than EA baselines.
>
> **Q2:** "Perhaps using ES is a good training strategy": Is the untrained block performing some sort of local or global mutation?
>
> **Answer 2:** "Perhaps using ES is a good training strategy" means that we are expected to implement a deeper BOptformer with the help of ES [1]. BOptformer currently suffers from difficulty in training deep architectures, so currently, BOptformer can perform fewer population iterations (depending on the number of layers of the architecture). However, we have experimentally observed an exciting phenomenon - deeper architectures can achieve better results. If we implement deep Optformer in future work, Optformer will achieve better results and become more general.
>
> The untrained BOptformer can perform mutation but is not necessarily suitable for the target black-box problem. In Table 4, for simple issues, the untrained BOptformer also achieves good results. But for complex planar mechanical arm problems, the results of untrained BOptformer are poor. The untrained BOptformer does not learn a mutation strategy adapted to the target task.
>
> [1] Paul Vicol, Luke Metz, and Jascha Sohl-Dickstein. Unbiased gradient estimation in unrolled computation graphs with persistent evolution strategies. In International Conference on Machine Learning, pp. 10553–10563. PMLR, 2021.

---

> ### Author Response · Authors · 2022-11-27
> **Did we clarify your comments?**
>
> We are very sorry to disturb you.
>
> Thanks also for your review of this article. Your review comments have made our manuscript a qualitative improvement. We don't know if our response clarified your query. We are also very much looking forward to your new insights into this article. We know you are very busy. However, we are very much looking forward to you taking the time out of your busy schedule to re-evaluate this article.
>
> Best wishes
>
> Authors

---

> > ### Comment · Reviewer_xa7p · 2022-12-02
> > **Increased score to 5**
> >
> > The paper is in a much better condition than before, and hence I raise my score to 5. It would significantly help if more visualization-based ablations were used to show what exactly the BOptFormer is doing. While Figure 3 is great, OB1 still raises skepticism: Did the BOptFormer really learn an intelligent behavior, or is it just spreading points randomly? Providing more evidence of the model's "intelligence" would go a long way.

---

> > > ### Author Response · Authors · 2022-12-09
> > > **More evidence of "intelligence"**
> > >
> > > We are very sorry to disturb you.
> > >
> > > We visualized the mutation strategy of BOptformer to explore its behavior. We use polynomial mutation, which is commonly used in evolutionary algorithms, as a reference. Given the input population (input), the mutated population (OB1) is obtained through OB1; the new population (mutpolyn) is obtained by performing polynomial mutation on the input population. We visualized the results of optimizing F4-F9. Sorry, we can no longer modify the original manuscript. The visualized results can be downloaded from the anonymous link below.
> > >
> > > https://anonymous.4open.science/r/ICLR-2023-Rebuttal-B753/vis_landscape.pdf
> > >
> > > We can observe the following phenomena:
> > >
> > > 1) The population generated by performing polynomial mutation is more evenly distributed on the landscape. However, most of the solutions produced by BOptformer's mutation strategy are concentrated in "areas with greater potential", which are closer to the optimal solution. Moreover, the population distribution generated by our scheme also takes diversity into account. In the non-optimal solution area, it is also more comprehensive than that of polynomial mutation, which is more conducive to jumping out of the local solution.
> > >
> > > 2) The population produced by performing polynomial mutation moves slightly compared to the original population. However, the mutation strategy of BOptformer can guide the input population to make big moves toward the optimal solution, which significantly accelerates the convergence of the algorithm.
> > >
> > > This shows that BOptformer is able to use the information of the objective function to guide the design of the mutation strategy, making it more applicable to the target optimization task, which is consistent with our motivation.
> > >
> > > These phenomena are very interesting. Due to the limited rebuttal time, we will do more analysis in the future to show the behavior of BOptformer. Thank you very much for your help.
> > >
> > > Best Wishes
> > >
> > > Authors

---

### Official Review · Reviewer_eoPi · 2022-10-20

**Confidence:** 5
**Correctness:** 1
**Technical Novelty And Significance:** 1
**Empirical Novelty And Significance:** 1
**Recommendation:** 3

**Clarity, Quality, Novelty And Reproducibility:**

Please see the comments above. I think this paper is not clearly written and its quality is poor. The scientific question is not clear and it is difficult to judge the novelty.

**Strength And Weaknesses:**

1. The English writing is poor. Too many long sentence making it difficult for understanding.
2. In the first paragraph, I do not think neural architecture search can be said that "we have no access to any other information about f". And it is not generally correct to say any hyperparameter optimization problems that "we have no access to any other information about f".
3. The second sentence, "evolutionary strategies" should be "evolution strategies". Also, Evolution strategies (ES) is a type of evolutionary algorithms(EAs), if the authors list EAs, it is no need to list ES.
4. What is "hand engineering"? And do not list 11 references at the same time.
5. "The rarely focus on black-box optimization problems", this is too strong and should be clearly justified. As they aim to solve general optimization problem, why cannot they solve black-box optimization problems.
6. "Although several efforts like Cao et al. (2019); Chen et al. (2017) have coped with these problems, they have limited performance due to poor representation ability of RNN, LSTM, and MLP." this is too strong and should be clearly justified.
7. The major motivation of this work is said as "Inspired by the similarity of EAs and Transformer", but there is no discussion about what is the similarity inbetween. And to the reviewer, EAs look not similar to Transformer.
8. For the experimental studies, the comparisons are too simple. First, the benchmark contains only six very simple functions. It is not enough to demonstrate the advantages of Optformer on these kinds of problems. Second, the real-world problem of Ab initio protein docking lacks of descriptions. It is unclear what the problem it is essentially. Third, the compared algorithm of DE look outdated as it is 12 years ago, and it is unclear what exact algorithms/versions of ES and CMA-ES are as there are references.



**Summary Of The Paper:**

This paper proposes a novel method for continuous unconstrained black-box optimization problems, called Optformer. It is claimed that the Optformer is inspired by the similarity between Transformer and EA. Six simple benchmark functions and a real-world problem are used to verify the performance of Optformer. In general, this paper is poorly written. I could not find the scientific question therein and there are many errors. I think the contribution of this work is rare.

**Summary Of The Review:**

In summary, I think this paper should be considerably re-organized and carefully improved in terms of at least motivation, related work, and experimental studies.

---

> ### Author Response · Authors · 2022-11-18
> **Response to Reviewer eoPi (Part 2)**
>
> **Q6:** "They rarely focus on black-box optimization problems", this is too strong and should be clearly justified. As they aim to solve general optimization problem, why cannot they solve black-box optimization problems.
>
> **Answer 6:** The listed papers require the use of gradient information of the objective function. However, the gradient information of the black box function is not available. Therefore, they cannot handle black-box optimization problems. Our statement is appropriate.
>
> **Q7:** "Although several efforts like Cao et al. (2019); Chen et al. (2017) have coped with these problems, they have limited performance due to poor representation ability of RNN, LSTM, and MLP." this is too strong and should be clearly justified.
>
> **Answer 7:** The comparison with L2O-Swarm (Cao (2019)) shows that our work achieves better results, therefore, we get this conclusion. We also make relevant corrections. At the same time, related research also shows that RNN, LSTM and MLP may not be as expressive as Transformer.
>
> **Q8:** The major motivation of this work is said as "Inspired by the similarity of EAs and Transformer", but there is no discussion about what is the similarity inbetween. And to the reviewer, EAs look not similar to Transformer.
>
> **Answer 8:** Reference [2] states the similarity between EA and transformer. While we've briefly covered this similarity in Chapter 4 of the text, we'll follow up with your comments for a more detailed introduction to this section.
>
> [2] Jiangning Zhang, Chao Xu, Jian Li, Wenzhou Chen, Yabiao Wang, Ying Tai, Shuo Chen, Chengjie Wang, Feiyue Huang, and Yong Liu. Analogous to evolutionary algorithm: Designing a unified sequence model., Advances in Neural Information Processing Systems, volume 34, pp. 26674–26688, 2021.
>
> **Q9:** For the experimental studies, the comparisons are too simple. First, the benchmark contains only six very simple functions. It is not enough to demonstrate the advantages of Optformer on these kinds of problems. Second, the real-world problem of Ab initio protein docking lacks of descriptions. It is unclear what the problem it is essentially. Third, the compared algorithm of DE look outdated as it is 12 years ago, and it is unclear what exact algorithms/versions of ES and CMA-ES are as there are references.
>
> **Answer 9:** For the Ab initio protein docking problem, we give the relevant citation [4] and introduce it in detail (see Section 4.1). We added a complex planar mechanic arm case to verify the effectiveness of the proposed algorithm further (see Section 4.2, Table 3). The experimental results are as follows:
> Simple Case (SC):  searching for different angles with fixed lengths. Complex Case (CC): searching for different angles and lengths.
> |  Case | $r$ | DE | ES | CMA-ES | L2O-Swarm | BOptformer | Untrained|
> | - | - | - | - | - | - | - | -|
> |SC | 100 | 1.20(0.64) | 10.6(5.58) | 1.36(0.35) | 40.4(3.89) | **0.30(0.18)** | 243(238)|
> |SC | 300 | 1.38(0.71) | 44.9(43.3) | 1.38(0.41) | 69.5(3.77) | **0.48(0.37)** | 1210(820)|
> |SC | 1000 | 93.8(137) | 183(239) | 43.7(110) | 176(7.20) | **26.6(57.4)** | 5070(2770)|
> |CC | 100 | 0.81(0.47) | 8.95(6.42) | 0.76(0.20) | 31.9(1.78) | **0.06(0.05)** | 243(238)|
> | CC | 300 | 6.15(12.2) | 47.8(56.0) | 0.87(0.37) | 89.1(1.96) | **0.50(0.79)** | 1210(820) |
> |CC | 1000 | 232(233) | 251(258) | 88.4(158) | 262(2.99) | **25.0(55.8)** | 5070(2770)|
>
> BOptformer is compared with standard EA baselines (DE(DE/rand/1/bin) [3], ES((\mu,\lambda)-ES), and CMA-ES), L2O-swarm[4] (a representative L2O method for black-box optimization), and Dragonfly [5]} (the state-of-the-art Bayesian optimization). DE and ES are implemented based on geatpy[6], and CMA-ES is implemented by pymoo[7]. These parameters are adjusted for each problem. We also added parameters for the comparison algorithms and descriptions of different experimental setups to make them reproducible.
>
> [3] Das, Swagatam, and Ponnuthurai Nagaratnam Suganthan. Differential Evolution: A Survey of the State-of-the-Art. IEEE Transactions on Evolutionary Computation, vol. 15, no. 1, IEEE, 2010, pp. 4-31.
>
> [4] Cao, Yue, et al. Learning to Optimize in Swarms. Advances in Neural Information Processing Systems, vol. 32, 2019, pp. 15044-54.
>
> [5] Kandasamy, Kirthevasan, et al. Tuning Hyperparameters without Grad Students: Scalable and Robust Bayesian Optimisation with Dragonfly. Journal of Machine Learning Research, vol. 21, no. 81, 2020, pp. 1-27.
>
> [6] Jazzbin, et. al. Geatpy: The Genetic and Evolutionary Algorithm Toolbox with High Performance in Python. 2020.
>
> [7] Blank, J., and K. Deb. Pymoo: Multi-Objective Optimization in Python. IEEE Access, vol. 8, 2020, pp. 89497-509.

---

> ### Author Response · Authors · 2022-11-18
> **Response to Reviewer eoPi (Part 1)**
>
> **Q1:** Please see the comments above. I think this paper is not clearly written and its quality is poor. The scientific question is not clear and it is difficult to judge the novelty. In summary, I think this paper should be considerably reorganized and carefully improved in terms of at least motivation, related work, and experimental studies.
>
> **Answer 1:** We have made corresponding changes. Please review our revised version for details. The blue fonts indicate the revised content. The modifications can be summarized as follows:
> 1) We renamed Optformer to BOptformer.
> 2) We remove the extensive introduction to evolutionary algorithms and VIT in Section 3. We focus on our proposed model and analyze experimental results.
> 3) Section 3.2 and Section 3.3 are moved to the Appendix.
> 5) We have redrawn the model architecture diagram (see Figure 1 in the revised version).
> 6) We add experimental results on the complex planner mechanic arm problem in Section 4.2 to further verify the effectiveness of our scheme.
> 7) We add a parametric analysis of the learning rate in Section 4.3 and show the results for different learning rates in the Appendix.
> 8) We visualize the learned SAC and FM strategies in Section 4.5.
> 9) The abbreviations in the experimental section have been written explicitly (see Section 4).
> 10) We revised the related work section to show our motivation (see Section 2).
> 11) We modified the description of the training set to make it clearer (see Section 3.6).
>
> **Motivation 1:** Although several efforts like {cao2019learning,chen2017learning} have coped with the black-box problems, their effectiveness may be hindered by the limited representational capabilities of RNN, LSTM, and MLP. We propose a solid Transformer-based L2O framework addressing black-box problems in the L2O community. We have demonstrated its benefit when compared with standard black-box optimization methods, particularly for the L2O-based method.
>
> **Motivation 2:** We require an expert to design or choose the evolutionary operations when given a new black-box optimization task to maximize its performance on the target task, which negatively impacts generalization ability. Most notably, the limited use of target function information in EA design due to expert knowledge limitations makes it difficult to adapt to the target task. The suggested BOptformer uses a Transformer framework instead of the manually designed crossover, mutation, and selection operators. The genetic operator is then designed automatically by the built Transformer rather than by a human designer. BOptformer efficiently uses the target black-box function's information to aid in developing the optimization strategy. In comparison to the human-designed EA, BOptformer has a substantially greater degree of task fit.
>
> **Q2:** The English writing is poor. Too many long sentence making it difficult for understanding.
>
> **Answer 2:** We have revised the full text according to your comments to make the article more accessible.
>
> **Q3:** In the first paragraph, I do not think neural architecture search can be said that "we have no access to any other information about f". And it is not generally correct to say any hyperparameter optimization problems that "we have no access to any other information about f".
>
> **Answer 3:** NAS and HPO are black-box optimization problems because their gradient and Hessian information cannot be obtained; only function estimation can be obtained. Please refer to the following reference.
>
> Daniel Golovin, et al. Google Vizier: A service for black-box optimization. In KDD, pages 1487–1495, 2017
>
> Of course, we also refer to this type of problem in particular. This description is also used in many pieces of literature. There are also efforts to build differentiable surrogate models of the original NAS and HPO problems to deal with. Still, we cannot change the conclusion that NAS and HPO are black-box optimization problems. We also welcome concrete examples from you.
>
> **Q4:** The second sentence, "evolutionary strategies" should be "evolution strategies".
>
> **Answer 4:** Fixed.
>
> **Q5:** What is "hand engineering"? And do not list 11 references at the same time.
>
> **Answer 5:** "Hand engineering" refers to when designing optimizers such as evolutionary algorithms or neural network optimizers (like Adam), we need to manually develop these optimizers based on expert knowledge. These optimizers designed according to expert knowledge may achieve good performance, which is irrelevant to the specific task. Therefore, the hand-designed optimizer will also perform differently on different optimization tasks. Consequently, we learn an optimizer that is more suitable for the target task according to some characteristics of the task itself. For a specific definition, refer to [1].
>
> As you suggested, we have fixed the problem of listing 11 references simultaneously.
>
> [1] Tianlong Chen, et al. Learning to optimize: A primer and a benchmark. JMLR, 23:1–59, 2022.

---

> > ### Comment · Reviewer_eoPi · 2022-12-02
> > **Thanks for the response**
> >
> > Thanks for the response. The authors have made their efforts to the rebuttal. I have carefully read the comments from other reviewers and the corresponding responses from the authors. I think some issues are clarified while the rest still confuses me.
> > 1) Reading from answer 3, I think the authors and me have different understanding of "black-box". The authors say that NAS and hyper-parameter optimization have no gradients, so they are black-box problems. However, I think they can be called non-derivative but not black-box. Because we do have the information about the architecture and the down-stream application of the neural networks.  Furthermore, hyper-parameter optimization is quite a general problem case that we cannot easily say we donot have information about it. Any model whose parameters cannot be self-tuned in its training process can be saied as hyper-parameters. And I even think NAS is a kind of hyper-parameter optimization, because the number of layers, the types of neurons and other architecture design units can be regarded and represented as the hyper-parameters of a network. In this regard, some claims about the black-box term in this paper may not convince me.
> >
> > 2) As this paper is motivated by the similarity of EA and Transformer, I still think it is necessary to demostrate how BOptformer is designed based on the  similarity and what essentially the similarity is. The answer 8 merely refers to a single paper without explaining the similarity. Of couse I can read that paper, but this makes this paper not self-contained because the similarity between EA and Transformer has not been a common sense in the research field and it is the major motivation of this work. Without knowing the similarity, I found myself difficult to judge the correctness of the technical design of BOptformer, because the technical descriptions are heavily based on the briefly concluded findings of the refered NeurIPS paper, and the reasons as well as the consideration behind those technical designs are largely omitted in this paper.
> >
> > 3) I still donot think the experiments are convincing enough to verify BOptformer. Would you please explain why training BOptformer on F1-F3 can make it successful on F4-F9? Are F1-F9 from the same distribution?  Without this justification, I think it is hard to tell whether the success on F4-F9 is due to the generalization of BOptformer.
> >
> > 4) What is the weakness of BOptformer comparing to traditional EAs?
> >
> > I do believe L2O is promising in its use and an End-to-End optimization model is interesting and worth investigation. But I found this paper still confuse me in the above aspects. So I decide to still keep my score unchanged, which is 3.
> >
> > Best regards

---

> > > ### Author Response · Authors · 2022-12-06
> > > **Response to New Comments (Part 2)**
> > >
> > > **Q3**: I still do not think the experiments are convincing enough to verify BOptformer. Would you please explain why training BOptformer on F1-F3 can make it successful on F4-F9? Are F1-F9 from the same distribution? Without this justification, I think it is hard to tell whether the success on F4-F9 is due to the generalization of BOptformer.
> > >
> > > **Answer 3**:
> > > **From the expression of the functions**, F1-F3 are low-fidelity surrogate functions of the target functions F4-F9. F1-F3 contain the information of objective functions (F4-F9). For example, F7 ($\sum \limits_{i=1}^{D} (z_i^2-10\cos(2\pi z_i)+10), z_i=x_i-b_i$), can be decomposed into $\sum \limits_{i=1}^{D} z_i^2-\sum \limits_{i=1}^{D}10\cos(2\pi z_i)+ \sum \limits_{i=1}^{D}10, z_i=x_i-b_i$. F2 is the low-fidelity surrogate function of $\sum \limits_{i=1}^{D} z_i^2$. F1 is the low-fidelity surrogate function of $\sum \limits_{i=1}^{D}10\cos(2\pi z_i)$. For other functions in F4-F9, we can find similar surrogate functions from F1-F3. BOptformer can use this information to maximize the matching degree between the learned optimization strategy and the objective function. However, F6 is less similar to F1-F3 than F4, F5, and F7-F9. Therefore, although the performance of BOptformer on F6 is better than that of the comparison algorithm, it is still poor.
> > >
> > > **From the perspective of landscape features**, F1-F3 include the following features: unimodal, multimodal, separable, and non-separable. The landscape features included in F4-F9 are as follows:
> > >
> > > F4: Unimodal, Separable
> > >
> > > F5: Unimodal, Separable
> > >
> > > F6: Multimodal, Non-separable, **Having a very narrow valley from local optimum to global optimum**
> > >
> > > F7: Multimodal, Separable, **Asymmetrical, Local optima’s number is huge**
> > >
> > > F8: Multi-modal, Non-separable, **Rotated**
> > >
> > > F9: Multi-modal, Non-separable, **Asymmetrical**
> > >
> > > The landscape features of F4 and F5 can be found in F1-F3. F6 has a new feature: "Having a very narrow valley from local optimum to global optimum". F7 has new features, "Asymmetrical" and "Local optima's number is huge". F8 has a new feature: "Rotated". F9 has a new feature, "Asymmetrical". The interference strength of different characteristics to landscape is arranged as follows: **Having a very narrow valley from local optimum to global optimum**>**Asymmetrical, Local optima’s number is huge**>**Asymmetrical**>**Rotated**. Therefore, BOptformer has the best generalization performance on F4, F5, and F8, the second-best generalization performance on F7 and F9, and the worst on F6.
> > >
> > > **Q4**: What is the weakness of BOptformer compared to traditional EAs?
> > >
> > > **Answer 4**:
> > > 1. Deep BOptformers are challenging to train, making it impossible to perform tens of thousands of iterations like EAs. Therefore, it may not be possible to obtain satisfactory results for very complex optimization scenarios.
> > > 2. BOptformer is less interpretable. There is no way to explain the learned optimization policy white box.

---

> > > ### Author Response · Authors · 2022-12-06
> > > **Response to New Comments (Part 1)**
> > >
> > > Thank you very much for your patient reply. We also very welcome your other questions.
> > >
> > > **Q1**: Reading from answer 3, I think the authors and me have a different understanding of "black-box".
> > >
> > > **Answer 1**: Thanks a lot for your guidance. Because some black-box optimization literature often uses NAS and hyperparameter optimization as the corresponding test cases. Therefore, we mistake them for black-box problems. After your guidance, we also feel that these statements are inappropriate. We will fix them in the modified version.
> > >
> > > **Q2**: The similarity of EA and Transformer.
> > >
> > > **Answer 2**: we are very sorry that the inappropriate statement has caused you trouble. In the modified version, we explain the similarity between EA and transformer (see Sections 3.2 and 3.3 in blue). The designed BOptformer is not a transformer architecture and its variants but a new, end-to-end L2O optimization framework. The similarity between EA and transformer should be re-expressed as the modified Transformer module can realize EA's crossover, mutation, and selection functions. That is, 1) We modified the MSA module to realize the information interaction between individuals in the population (similar to EA’s crossover operation); 2) we modified the FFN module to realize the individual’s mutation evolution (similar to EA’s mutation operation); 3) we modified the residual module to achieve the survival of the fittest (similar to the selection operation of EA). Finally, BOptformer organically combines these modules to realize the mapping from the initial population to the optimal solution (similar to EA's population iteration process).
> > >
> > > At the same time, we summarize the similarities between the two as follows:
> > >
> > > **1. Crossover operator vs. SA**
> > >
> > > The crossover operator generates a new individual by $\sum_{i=1}^n X_i W_i^c$. $W_i^c$ is the diagonal matrix and $X_i$ is one individual.
> > >
> > > For the SA module in the transformer (MSA degenerates to SA when there is only one head), each patch embedding interacts with all embeddings. The head feature $H_i$ can be formulated as:
> > > $$
> > > H_i = SA(QW_i^Q, KW_i^K, VW_i^V) = Softmax\left(QW_i^Q(KW_i^K)^T/sqrt(d_k)\right)VW_i^V = AVW_i^V
> > > $$
> > > where $W_i^Q \in R^{d_m \times d_q}$, $W_i^K \in R^{d_m \times d_k}$, and $W_i^V \in R^{d_m \times d_v}$ are parameter matrices for queries, keys, and values, respectively; $W^O \in R^{hd_v \times d_m}$ maps each head feature $H_i$ to the output. Moreover, $d_m$ is the input dimension, while $d_q$, $d_k$, and $d_v$ are hidden dimensions of the corresponding projection subspace; $h$ is the head number. $A \in R^{l \times l}$ is the attention matrix of $h$th head, $l$ is the sequence length.
> > >
> > > **By comparing $AVW_i^V$ with $\sum_{i=1}^n X_i W_i^c$, we find that they share the same formula representation.**
> > >
> > > **2. Mutation operator vs. FFN**
> > >
> > > The mutation operator brings random changes into the population. Specifically, an individual $X_i$ in the population goes through the mutation operator to form the new individual $\hat{X}_i$, formulated as $\hat{X}_i = X_i W_i^m$. $W_i^m$ is the diagonal matrix.
> > >
> > > FFN employs two cascaded linear transformations with a ReLU activation to handle $X$, which is shown as $FFN(X) = max(0, XW_1 + b_1)W_2 + b_2$, where $W_1$ and $W_2$ are weights of two linear layers, and $b_1$ and $b_2$ are corresponding biases.
> > >
> > > Take one-layer linear as an example: $X_i = W_1 X_i$, where $ W_1$ is the weight of the first linear layer of the FFN module, and it is applied to each position separately and identically.
> > >
> > > **By comparing $\hat{X}_i = X_i W_i^m$ with $X_i = W_1 X_i$, we find that they share the same formula representation.**
> > >
> > > **3. Selection Operator vs. Residual Connection.**
> > >
> > > In the evolution of the biological population, individuals at the previous stage have a certain probability of inheriting the next stage. This phenomenon is expressed in the transformer as a residual connection, i.e., patch embeddings of the previous layer are directly mapped to the next layer.

---

> ### Author Response · Authors · 2022-11-27
> **Did we clarify your comments?**
>
> We are very sorry to disturb you.
>
> Thanks also for your review of this article. We don't know if our response clarified your query. We are also very much looking forward to your new insights into this article. We know you are very busy. However, we are very much looking forward to you taking the time out of your busy schedule to re-evaluate this article.
>
> Best wishes
>
> Authors

---

### Official Review · Reviewer_qcoV · 2022-10-22

**Confidence:** 3
**Correctness:** 3
**Technical Novelty And Significance:** 3
**Empirical Novelty And Significance:** 3
**Recommendation:** 6

**Clarity, Quality, Novelty And Reproducibility:**

The paper writing is generally clear and easy to understand. There are only some parts that I don't feel clear enough and need to seek further information - please see my comments above.

The key idea of the paper is interesting and novel. The proposed method seems to be sound to me. The experiments show that the proposed method performs well on various problems (but note there is only one real-world problem) - please see my comments above.

The paper provides the code which can help the reproducibility.

**Strength And Weaknesses:**

Strengths:

Overall, I think the key idea of the paper, which is to propose a transformer-based model to solve the unconstrained black-box optimization problems is very interesting and novel.
+ The key idea of using the similarity between the Vision Transformer and evolutionary algorithms (EAs), and then developing a new model by modifying the Transformer's components to implement the EA operations seems to be sound and novel to me.
+ The paper's writing is generally clear and easy to understand, although there are some parts that I feel I need to understand more (please see my questions below).
+ The experimental evaluation shows the strong performance of the proposed method. It also includes some ablation study and sensitivity analysis to further understand the behavior of the proposed method.

Weaknesses:
+ Using a very deep learning model to solve the black-box optimization problems will introduce a lot of extra hyper-parameters that may significantly affect the performance of the proposed method. I found it hard to believe that an untrained model could still yield good results. Perhaps the test problems are not complex enough?
+ The experimental evaluation only includes one real-world problem (protein docking). For these types of work, I expect more real-world problems to be included in the experimental evaluation.

Questions:
+ In Section 4.5, I feel very unclear about how the training dataset is generated. The paper states that the method to generate the training dataset is to "randomly produce a shifted objective function by adjusting the location of the optima", but we do not know the objective functions (they're black-box functions) so how can we generate these shifted objective functions? In the experimental evaluation, I notice that the paper uses 3 new functions to create the training datasets, so how are these functions chosen? Do they need to share the similarity with the objective functions that we need to optimize?
+ For the loss function in Eq. (12), a simple model that minimizes this loss function is the model where E_{\theta}(X0) = X(0), but if this happens, then we can never find the optimal solution set of the black-box optimization problem. Will this case ever happen for the proposed method?
+ The paper always states that X0 needs to be sorted in non-descending order of fitness (e.g., in the first sentence of page 6), but it never describes how we sort this set X0. Does the paper mean to say that the proposed Optformer will actually perform this ordering of fitness?
+ In the experiments, the number of generations of all the reference algorithms is set to 100. This seems to be quite a small value given that some objective functions have the dimensions of 100. Plots showing the optimal values found by all the algorithms w.r.t. the number of generations (and the number of generations is much larger than 100) might be helpful.
+ Is there any guideline on how to choose the hyper-parameters of the Optformer? E.g., learning rate, number of layers, etc.


**Summary Of The Paper:**

The paper proposes a new approach based on Transformer, namely Optformer, to optimize unconstrained black-box optimization problems. The key idea is based on an existing observation that Vision Transformer is similar to the evolutionary algorithm (EAs). Therefore, the paper aims to modify the components of the Transformer to implement the evolutionary algorithm's operations such as crossover, mutation, and selection. The proposed model Optformer then creates a mapping between a random population and the optimal solution of the black-box optimization problem. Experiments are conducted on 6 black-box functions and one real-world problem to show the efficacy of the proposed method.

**Summary Of The Review:**

Overall, I think the key idea of the paper, which is to propose a transformer-based model to solve the unconstrained black-box optimization problems is very interesting and novel. The proposed method is sound to me. It is also shown to perform well on various problems. The paper's writing is generally clear. But I still have various concerns as listed in my questions. If these concerns are addressed, I can increase the score for the paper.

---

> ### Author Response · Authors · 2022-11-18
> **Response to Reviewer qcoV (Part 3)**
>
> **Q7:** In the experiments, the number of generations of all the reference algorithms is set to 100. This seems to be quite a small value given that some objective functions have the dimensions of 100. Plots showing the optimal values found by all the algorithms w.r.t. the number of generations (and the number of generations is much larger than 100) might be helpful.
>
> **Answer 7:** In the optimization process, BOptformer only iterates a few generations. For example, Optformer (nws5) has only evolved five generations, and Optformer (ws30) has only evolved 30 generations. To ensure the fairness of the experiment, we should also set the maximum evolutionary generation of EA baselines to 5 generations (30 generations). However, we uniformly set the iteration number of EA baselines to 100 generations. EA baselines have 100/5 times as many function evaluations as Optformer (nws5). Although this setting is not good for Optformer, the experimental results show that BOptformer can still achieve much better results than baselines. Such experimental results are encouraging.
>
> Meanwhile, we have observed that a deep BOptformer achieves better results. However, the vanishing gradient problem makes deep Optformers challenging to train. If we can design an optimization method to train a deeper BOptformer, then the Optformer will have a stronger search ability. Probably an evolution strategy is a good solution [2].
> We plotted the convergence curves of BOptformer (ws10), ES, DE, and CMA-ES on F7 (see Appendix A.7, Figure 5). BOptformer converges quickly and can obtain better solutions. BOptformer can only iterate ten times to get the best solution relative to EA baselines. ES and DE converged around 100 generations, and CMA-ES showed a slow convergence rate.
>
> [2]Vicol, Paul, Luke Metz, και Jascha Sohl-Dickstein. Unbiased gradient estimation in unrolled computation graphs with persistent evolution strategies. International Conference on Machine Learning. PMLR, 2021. 10553–10563.
>
> **Q8:** Is there any guideline on how to choose the hyper-parameters of the Optformer? E.g., learning rate, number of layers, etc.
>
> **Answer 8:** **1. Learning rate**. We have added a new parametric analysis of the learning rate (see Section 4.3), and the experimental results are shown in Table 9 (Appendix A.6). We can draw the following conclusions:
>
> 1) BOptformer-nws5 and BOptformer-ws30 perform poorly when the learning rate is 0.1, which may affect the convergence of the model during the training process because the learning rate is too large.
>
> 2) For BOptformer-nws5, setting the learning rate to 0.01 achieves relatively good performance.
>
> 3) For BOptformer-ws30 and BOptformer-ws3, lr=0.0001 is a good choice.
>
> **2. Structure**. For simple problems, you can choose an architecture with fewer layers and weight sharing, such as BOptformer (ws3). For general problems, you can choose an architecture with many layers and weight sharing, BOptformer (ws30). For very complex problems, you can choose an architecture with a large number of layers and no weight sharing, such as BOptformer (n15). However, the architecture cannot be too deep, making training difficult due to vanishing gradients.
> In the future, we will use automatic machine learning methods to search for better hyperparameters and architectures to improve the performance of BOptformer further. We will also further explore ways to train deeper architectures.
> |BOptformer(nws5)|
> |-|
>
> | $lr$    | F4    | F5    | F6    | F7    | F8    | F9|
> | - | -   | -  | - | - | - | -|
> |0.1   | 0.93(7.42) | 0.31(0.49) | 2.04e7(2.03e8) | 15.3(6.4) | 0.36(0.16) | 0.61(0.18)|
> |0.01  | **0.01(0.003)** | **0.05(0.02)** | **9.57(0.22)**| **1.62(0.60)** | **0.03(0.01)** | **0.06(0.03)**|
> |0.001 | 0.88(3.41) | 0.36(0.12) | 226(1750) | 6.18(2.66) | 0.56(0.16) | 1.36(0.36) |
> |0.0001 | 0.06(0.03) | 0.13(0.03) | 13.6(2.11) | 0.83(0.50) | 0.17(0.04) | 0.28(0.10)|
>
> |BOptformer(ws30)|
> |-|
>
> | $lr$    | F4    | F5    | F6    | F7    | F8    | F9|
> | - | -   | -  | - | - | - | -|
> | 0.1   | 1.64(1.19) | 0.85(1.59) | 493(3110) | 28.4(4.35) | 0.47(0.11) | 2.82(0.5) |
> | 0.01  | 0.05(0.30) | 0.09(0.03) | 39.2(240) | 1.05(1.40) | 0.01(0.06) | 0.28(0.07)|
> |0.001 | **1.01e-3(0.001)** | 0.02(0.01) | 9.03(0.18) | 0.03(0.02) | **0.003(0.001)** | 0.03(0.01)|
> |0.0001 | 1.50e-3(0.001) | **0.016(0.004)** | **9.01(0.13)** | **0.02(0.02)** | 0.006(0.002) | **0.02(0.0.01)** |
>
>  |BOptformer(ws3)|
> |-|
>
> | $lr$    | F4    | F5    | F6    | F7    | F8    | F9|
> | - | -   | -  | - | - | - | -|
> | 0.1   | 3.04(5.55) | 0.98(0.4) | 1150(7930) | 43.3(8.87) | **0.64(0.12)** | 2.43(0.98) |
> |0.01  | 29.9(47.7) | 2.68(1.30) | 6.24e4(4.87e5) | 40.4(8.52) | 1.05(0.06) | 4.45(0.85) |
> |0.001 | 1.82(1.20) | 0.76(0.32) | 654(4780) | 7.00(7.11) | 0.79(0.13) | 1.91(0.53) |
> |0.0001 | **0.39(0.21)** | **0.33(0.07)** | **46.8(79.4)**| **2.22(2.41)** | 0.66(0.09) | **0.59(0.19)**|

---

> ### Author Response · Authors · 2022-11-18
> **Response to Reviewer qcoV (Part 2)**
>
> **Q3:** The experimental evaluation only includes one real-world problem (protein docking). For these types of work, I expect more real-world problems to be included in the experimental evaluation.
>
> **Answer 3:** We have added a complex planar mechanic arm problem (see Section 4.2) and experimental results in Answer 2. BOptformer has a significant advantage.
>
> **Q4:** In Section 4.5, I feel very unclear about how the training dataset is generated. The paper states that the method to generate the training dataset is to "randomly produce a shifted objective function by adjusting the location of the optima", but we do not know the objective functions (they're black-box functions) so how can we generate these shifted objective functions? In the experimental evaluation, I notice that the paper uses 3 new functions to create the training datasets, so how are these functions chosen? Do they need to share the similarity with the objective functions that we need to optimize?
>
>
> **Answer 4:** The functions in the training dataset we construct are differentiable surrogate functions for the target black-box problem. We get information about these differentiable surrogate functions so that we can perturb these surrogate functions. We can also build these surrogate functions via UCB in Bayesian optimization.
>
> In the Ab initio protein docking problem and the planner mechanic arm problem, the constructed surrogate function set is of high fidelity, and our scheme achieves good performance. We often struggle to obtain high-fidelity differentiable surrogate functions. The constructed cheap differentiable surrogate is somewhat different from the target black-box problem, called the low-fidelity training set. For example, F1-F3 and F4-F9, these two problem sets are quite different. Therefore, F1-F3 are low-fidelity surrogate function sets of F4-F9. Nevertheless, we still observe that the BOptformer trained on F1-F3 can still perform well on the target black-box problem.
>
> **We rewrote the training set section (see Section 3.6) as follows:**
>
> “Before introducing the details of the training dataset, fidelity [1] is defined as follows:
>
> Suppose the differentiable surrogate functions $f_1, f_2, \cdots, f_m$ are the continuous exact approximations of the black-box function $f$. We call these approximations fidelity, which satisfies the following conditions:
>
> 1) $f_1, \cdots, f_i, \cdots, f_m$ approximate $f$. $||f-f_i||_{\infty} \leq \zeta_m$, where the fidelity bound $\zeta_1>\zeta_2> \cdots \zeta_m$.
>
> 2) Estimating approximation $f_i$ is cheaper than estimating $f$. Suppose the query cost at fidelity is $\lambda_i$, and $\lambda_1<\lambda_2< \cdots \lambda_m$.}
>
> Training data is a crucial factor beyond the objective functions. This paper establishes the training set by constructing a set of differentiable functions related to the optimization objective.
> This training dataset only contains $(X_0, f_i(x|\omega))$, the initial population and objective function, respectively. The variance of $\omega$ causes the shift in landscapes. The training dataset is designed as follows: 1) Randomly initialize the input population $X_0$; 2) Randomly produce a shifted objective function $f_i(x|\omega)$ by adjusting the parameter $\omega$; 3) Evaluate $X_0$ by $f_i(x|\omega)$; 4) Repeat Steps 1)-3) to generate the corresponding dataset. We show the designed training and testing datasets as follows:
> $$
> F^{train} = \{ f_1(x|\omega_{1,i}^{train}),\cdots, f_m(x|\omega_{m,i}^{train})\}
> $$
> where $\omega_{m,i}^{train}$ represents the $i$th different values of $\omega$ in $m$th function $f_m$.”
>
> [1] Kandasamy, Kirthevasan, et al. “Gaussian Process Bandit Optimisation with Multi-Fidelity Evaluations.” NIPS, 2016.
>
> **Q5:** For the loss function in Eq. (12), a simple model that minimizes this loss function is the model where E_{\theta}(X0) = X(0), but if this happens, then we can never find the optimal solution set of the black-box optimization problem. Will this case ever happen for the proposed method?
>
> **Answer 5:** Our scheme implements the mapping from the initial population to the target population. The elite retention strategy is implemented in the selection module. If this happens, the optimal solution is preserved. Moreover, the function value of formula Eq. (12) is 0 at this time, which means that our scheme has converged to a good state, and no more training is needed.
>
> **Q6:** The paper always states that X0 needs to be sorted in non-descending order of fitness (e.g., in the first sentence of page 6), but it never describes how we sort this set X0. Does the paper mean to say that the proposed Optformer will actually perform this ordering of fitness?
>
> **Answer 6:** We use quicksort to sort the population.

---

> ### Author Response · Authors · 2022-11-18
> **Response to Reviewer qcoV (Part 1)**
>
> Thanks very much for your positive review of our article. Your comments make our article a qualitative improvement. We have also uploaded the revised version, and we kindly ask you to re-evaluate our paper.
>
> **Q1:** Using a very deep learning model to solve the black-box optimization problems will introduce a lot of extra hyper-parameters that may significantly affect the performance of the proposed method.
>
> **Answer 1:** The hyperparameters of BOptformer are the learning rate $lr$, whether the weights share, and the number of OBs. These are the parameters that mainly affect the performance of BOptformer. We present the corresponding results in the parametric analysis section (see Section 4.3). Good performance has been achieved with the default settings of MSA and FFN. Here, we do not discuss. Of course, if an automatic machine learning method is introduced to search for hyperparameters and model architectures, it is expected to improve the performance of BOptformer further.
>
> **Q2:** I found it hard to believe that an untrained model could still yield good results. Perhaps the test problems are not complex enough?
>
> **Answer 2:** The untrained model can perform well on the six problems with d = 10 in Table 7. These questions are somewhat simple. Untrained models have the ability to generate and select solutions and also have certain optimization capabilities. But in complex problems, this conclusion does not hold. We have also revised the statement in this section. At the same time, we added experiments on the complex planar mechanic arm problem (see Table 3, Section 4.2) and tested the performance of the untrained model on this problem. We found that the untrained model could not achieve satisfactory results. The experimental results are shown as follows. Untrained represents the untrained BOptformer.
>
> | Case | $r$ | DE  | ES | CMA-ES | L2O-Swarm | BOptformer | Untrained|
> | -| - | - | - | -| - | - | -|
> |SC | 100 | 1.20(0.64) | 10.6(5.58) | 1.36(0.35) | 40.4(3.89) | **0.30(0.18)** | 243(238) |
> |SC | 300 | 1.38(0.71) | 44.9(43.3) | 1.38(0.41) | 69.5(3.77) | **0.48(0.37)** | 1210(820) |
> |SC| 1000 | 93.8(137) | 183(239) | 43.7(110) | 176(7.20) | **26.6(57.4)** | 5070(2770) |
> |CC | 100 | 0.81(0.47) | 8.95(6.42) | 0.76(0.20) | 31.9(1.78) | **0.06(0.05)** | 243(238) |
> |CC | 300 | 6.15(12.2) | 47.8(56.0) | 0.87(0.37) | 89.1(1.96) | **0.50(0.79)** | 1210(820) |
> |CC | 1000  | 232(233) | 251(258) | 88.4(158) | 262(2.99) | **25.0(55.8)** | 5070(2770)|

---

> ### Author Response · Authors · 2022-11-27
> **Did we clarify your comments?**
>
> We are very sorry to disturb you.
>
> Thanks also for your review of this article. Your review comments have made our manuscript a qualitative improvement. We don't know if our response clarified your query. We are also very much looking forward to your new insights into this article. We know you are very busy. However, we are very much looking forward to you taking the time out of your busy schedule to re-evaluate this article.
>
> Best wishes
>
> Authors

---

> > ### Comment · Reviewer_qcoV · 2022-12-01
> > **Response after rebuttal**
> >
> > Dear authors,
> >
> > Sorry for the late response. I have gone through your response and some of my concerns have been addressed but some still remain. In particular, I still think that the extra hyper-parameters caused by the use of a very deep learning model will affect the performance of the proposed method. Good performance we see in the results is only based on several problems (and only two real-world problems) so I don't think we can quantify the effect of these extra hyper-parameters yet. More thorough evaluation needs to be conducted in order to understand the performance of the proposed method. For this, I decided to still keep my score as the original score before rebuttal, which is 6.
> >
> > Warm regards,

---

> > > ### Author Response · Authors · 2022-12-05
> > > **Response to New Comments**
> > >
> > > Many thanks for your recognition of our work.
> > >
> > > First of all, we are very sorry and hope you understand. After our testing and thinking, we decided not to do more experiments to show the effect of hyperparameters. **We believe that the purpose of parameter analysis is to provide guidance for new tasks. Even if we do other experiments, it is difficult for us to get other useful knowledge.**
> > >
> > > Second, we agree with you. The extra hyperparameters caused by using a deep learning model will affect the performance of our proposal. Our experimental results also show the existence of this effect. We do not think it makes sense to show the effect of hyperparameters on other problems. With different target tasks, the optimal parameter combination is always different. We cannot give a unified answer even if we test on more problems.
> > >
> > > The hyperparameters of BOptformer mainly include the following two aspects:
> > >
> > > 1) Adam's learning rate. The choice of learning rate determines whether BOptformer can be trained appropriately. In the face of different tasks, we cannot directly give the appropriate learning rate. We think you are aware that this is difficult. We can only find a suitable one from the recommended learning rate set.
> > >
> > > 2) The structure of BOptformer. This includes two aspects: the number of layers of BOpformer (the number of OBs) and whether to implement weight sharing between OBs. Even without experimental verification, we can give the following recommendations. This is because the reason for doing this is very intuitive.
> > > Similar to other neural networks, deeper architecture and non-weight sharing between layers can always enhance the representation ability of BOptformer, but the training difficulty is also increasing. As the depth increases, the performance of BOptformer increases first and then decreases. Different task characteristics will make it challenging to give the optimal number of layers in advance. Nor does testing on new problems change this conclusion. Perhaps we can count on AutoML to solve this problem. In order to balance computational complexity and performance, we give the following suggestions:
> > >
> > > **For simple problems, we can choose an architecture with fewer layers and weight sharing. We can choose an architecture with many layers and weight sharing for general problems. For very complex problems, we can choose an architecture with a large number of layers and no weight sharing. However, the architecture cannot be too deep, making training difficult due to vanishing gradients.**
> > >
> > > We hope this answer will clear your concerns. We also welcome your other inquiries.

---

### Official Review · Reviewer_YunP · 2022-10-24

**Confidence:** 5
**Clarity, Quality, Novelty And Reproducibility:** Please see above.
**Correctness:** 3
**Technical Novelty And Significance:** 3
**Empirical Novelty And Significance:** 2
**Recommendation:** 5

**Strength And Weaknesses:**

Strengths:
- Interesting approach with potential impact on black-box optimisation
- Favourable results in application domains considered

Weaknesses:

- Paper writing: I have found the paper hard to follow and not clearly written. Of course, I feel with the authors who had to introduce many related concepts before proposing their method. Maybe the authors could consider moving their contributions before page 4 to make it easier to grasp the novelty of the paper.

- There does not seem to be regard to sample efficiency in the proposed approach. Are the authors not worried about expensive of evaluating black-box functions? I ask since the domains considered in the experiments are relatively toy and running the experiments in a real-world problem beyond Ab initio docking. Looking at the literature on high-dimensional Bayesian optimisation (which needs to be cited) can help in designing such experiments.

- I am also a bit confused about the experimental results, which seem to somewhat contradict the notion of black-box optimisation. To elaborate, it seems that the training set's similarity to the black box is a critical factor in the approach's success. If that is the case, how can we define this notion? If the target task (so to say) is a black box then what does it mean for us to be able to pick similar training data? What does similarity mean in this context?

- Why haven't the authors compared to state-of-the-art bayesian optimisation solutions in their experiments? As far as I can tell, the dimensions vary from 10 to 100 d which SOTA BO should be able to handle.

- What motivated the choices of the training and testing functions? After reading the paper, they seemed to be arbitrary. Maybe the authors can help me understand?

**Summary Of The Paper:**

In this paper, the authors proposed an interesting approach based on what I can tell to be a novel connection between vision transformers and evolutionary algorithms. The paper then modifies the transformer architecture to implement evolutionary-specific operations like crossover, mutation and selection. The authors then demonstrate their method on six black-box functions and show superior performance to baselines.

**Summary Of The Review:**

In general, I think this is an interesting approach. In its current state, however, I can not recommend acceptance. I am of course willing to change my score if the authors convince me in the rebuttal phase.

---

> ### Author Response · Authors · 2022-11-18
> **Response to Reviewer YunP (Part 2)**
>
> **Q3:** I am also a bit confused about the experimental results, which seem to somewhat contradict the notion of black-box optimisation. To elaborate, it seems that the training set's similarity to the black box is a critical factor in the approach's success. If that is the case, how can we define this notion? If the target task (so to say) is a black box then what does it mean for us to be able to pick similar training data? What does similarity mean in this context?
>
> **Answer 3:**  Similarity refers to fidelity [6], which is defined as follows:
> Suppose $f_1, f_2, \cdots, f_m$ are the continuous exact approximations of the objective function $f$. We call these approximations fidelity, which satisfies the following conditions:
>
> 1) $f_1, \cdots, f_i, \cdots, f_m$ approximate $f$. $||f-f_i||_{\infty} \leq \zeta_m$, where the fidelity bound $\zeta_1>\zeta_2> \cdots \zeta_m$.
>
> 2) Estimating approximation $f_i$ is cheaper than estimating $f$. Suppose the query cost at fidelity is $\lambda_i$, and $\lambda_1<\lambda_2< \cdots \lambda_m$.
>
> The training set is the differentiable surrogate function we construct for the target black-box problem, and the closer the differentiable surrogate is to the target black-box problem, the more promising we can get a better solution to the target black-box problem. We can also build these surrogate functions through UCB. In the Ab initio protein docking problem and the planner mechanic arm problem, the constructed surrogate function set is of high fidelity, and our scheme achieves good performance.
>
> We often struggle to obtain high-fidelity differentiable surrogate functions. The constructed cheap differentiable surrogate is somewhat different from the target black-box problem, called the low-fidelity training set. For example, F1-F3 and F4-F9, these two problem sets are quite different. Therefore, F1-F3 are low-fidelity surrogate function sets of F4-F9. Nevertheless, we still observe that the BOptformer trained on F1-F3 can still perform well on the target black-box problem.
>
> [6] Kandasamy, Kirthevasan, et al. “Gaussian Process Bandit Optimisation with Multi-Fidelity Evaluations.” NIPS, 2016.
>
> **Q4:** Why haven't the authors compared to state-of-the-art bayesian optimisation solutions in their experiments? As far as I can tell, the dimensions vary from 10 to 100 d which SOTA BO should be able to handle.
>
> **Answer 4:** Dragonfly [1] in Table 1 is a state-of-the-art Bayesian optimization. On the 100-dimensional problem, Dragonfly needs dozens of hours to complete the search, and the obtained solution is poor. The time it takes to find the best hyperparameters is even more outrageous. our scheme can achieve better results within 1s. The results of Dragonfly on the 100-dimensional problems are shown as follows:
>
> | $f$ | Dragonfly | BOptformer|
> | - | - | -|
> | F4 | 11200(3750) | **0.11(0.09)**|
> | F5 | 50(0) | **0.14(0.15)**|
>  | F6 | **99(0)** | 129(346) |
> | F7 | 144(13.1) | **24.1(13)**|
> | F8  | 125(11.3) | **0.02(0.03)**|
> | F9 | 10.5(0.32) | **0.15(0.05)**|
>
> **Q5:**: What motivated the choices of the training and testing functions? After reading the paper, they seemed to be arbitrary. Maybe the authors can help me understand?
>
> **Answer 5:** Test functions are the black-box problem we need to solve. The training function is a differentiable surrogate function built for the target black-box problem and used to train the parameters in BOptformer. The constructed differentiable surrogate problem has different fidelity than the target black-box problem. See Answer 3 for the definition of fidelity.

---

> ### Author Response · Authors · 2022-11-18
> **Response to Reviewer YunP (Part 1)**
>
> Thanks very much for your positive review of our article. Your comments make our article a qualitative improvement. We have also uploaded the revised version, and we kindly ask you to re-evaluate our paper.
>
> **Q1:** Paper writing: I have found the paper hard to follow and not clearly written. Of course, I feel with the authors who had to introduce many related concepts before proposing their method. Maybe the authors could consider moving their contributions before page 4 to make it easier to grasp the novelty of the paper.
>
> **Answer 1:** Thank you for your comments. We have made corresponding changes. Please review our revised version for details. The blue fonts indicate the revised content. The modifications can be summarized as follows:
>
> 1) We renamed Optformer to BOptformer.
>
> 2) We remove the extensive introduction to evolutionary algorithms and VIT in Section 3. We focus on our proposed model and analyze experimental results.
>
> 3) Section 3.2 and Section 3.3 are moved to the Appendix.
>
> 5) We have redrawn the model architecture diagram (see Figure 1 in the revised version).
>
> 6) We add experimental results on the complex planner mechanic arm problem in Section 4.2 to further verify the effectiveness of our scheme.
>
> 7) We add a parametric analysis of the learning rate in Section 4.3 and show the results for different learning rates in the Appendix.
>
> 8) We visualize the learned crossover and mutation strategies in Section 4.5.
>
> 9) The abbreviations in the experimental section have been written explicitly (see Section 4).
>
> **Q2:** There does not seem to be regard to sample efficiency in the proposed approach. Are the authors not worried about expensive of evaluating black-box functions? I ask since the domains considered in the experiments are relatively toy and running the experiments in a real-world problem beyond Ab initio docking. Looking at the literature on high-dimensional Bayesian optimisation (which needs to be cited) can help in designing such experiments.
>
> **Answer 2:** During the training phase, we construct a set of cheap differentiable surrogate functions for the target's black-box function and then train BOptformer on these differentiable surrogate functions. Therefore, directly estimating the target black-box function during the training phase is unnecessary. Of course, we can also use the UCB method in Bayesian optimization to assist in the construction of the training set. In the testing phase, we use the trained BOptformer to optimize the target black-box function. Unlike Bayesian optimization, which requires thousands of iterations to end the search, our scheme can achieve good solutions when the population iterates only a few times (the specific number of population iterations depends on the architecture used), and the convergence speed is breakneck. Moreover, the optimization process can be performed very quickly with the support of the GPU. Therefore, we are not worried about the expensive evaluation of black-box functions.
>
> We added a more challenging experiment with the planar Mechanic Arm (see Section 4.1). Experimental results with other baselines demonstrate the effectiveness and efficiency of our scheme.
>
> We also cite the related method of high-dimensional Bayesian optimization [1]-[5].
>
> [1] Kandasamy, Kirthevasan, et al. Tuning Hyperparameters without Grad Students: Scalable and Robust Bayesian Optimisation with Dragonfly. Journal of Machine Learning Research, vol. 21, no. 81, 2020, pp. 1-27.
>
> [2] Mojmir Mutny and Andreas Krause. Efficient high dimensional bayesian optimization with additivity and quadrature fourier features, Advances in Neural Information Processing Systems, volume 31. Curran Associates, Inc., 2018.
>
> [3] Cheng Li, Sunil Gupta, Santu Rana, Vu Nguyen, Svetha Venkatesh, and Alistair Shilton. High dimensional bayesian optimization using dropout. IJCAI’17, 2017.
>
> [4] Kirthevasan Kandasamy, Jeff Schneider, and Barnab as Poczos. High dimensional bayesian optimisation and bandits via additive models. In International conference on machine learning, pp. 295–304. PMLR, 2015.
>
> [5] Maximilian Balandat, Brian Karrer, Daniel Jiang, Samuel Daulton, Ben Letham, Andrew G Wilson, and Eytan Bakshy. Botorch: a framework for efficient monte-carlo bayesian optimization. Advances in neural information processing systems, 33:21524–21538, 2020.

---

> ### Author Response · Authors · 2022-11-27
> **Did we clarify your comments?**
>
> We are very sorry to disturb you.
>
> Thanks also for your review of this article. Your review comments have made our manuscript a qualitative improvement. We don't know if our response clarified your query. We are also very much looking forward to your new insights into this article. We know you are very busy. However, we are very much looking forward to you taking the time out of your busy schedule to re-evaluate this article.
>
> Best wishes
>
> Authors

---

> > ### Author Response · Authors · 2022-12-06
> > **Did we clarify your comments？(2)**
> >
> > We are very sorry to bother you. We know you are very busy. We don't know if our reply clears up your doubts. We also very much look forward to discussing further with you.
> >
> > Best Wishes
> >
> > Authors

---

> > > ### Comment · Reviewer_YunP · 2022-12-06
> > > **Thank you**
> > >
> > > Dear Authors,
> > >
> > > Sorry for the late reply.
> > >
> > > I went through the responses and plan to increase my score. Some comments have been addressed. I am still not convinced of the proposed technique's black box and sample-efficient nature. Does the pre-training phase require domain knowledge of the black-box we want to optimise? If so what is this knowledge and what are those assumptions?
> > >
> > > I thank you for the clarifications and extra experiments. I also think the diagram is very useful.

---

> > > > ### Author Response · Authors · 2022-12-06
> > > > **Response to New Comments**
> > > >
> > > > Thank you so much for raising your score. We hope the following reply will clear your doubts.
> > > >
> > > > **Q1**: I am still not convinced of the proposed technique's black box and sample-efficient nature.
> > > >
> > > > **Answer1** :
> > > >
> > > > **Unlock the black box of BOptformer**
> > > >
> > > > In order to eliminate the black box of BOptformer, we visualize and analyze the policies learned by the OB layer of BOptformer (see Section 4.5, Figures 2 and 3). We are very sorry because we need to trouble you to review our revised version as figures cannot be uploaded here.
> > > >
> > > > Figure 2 shows the crossover strategies learned by the 5 OBs (actually SACs) of BOptformer. The changes offered by the crossover strategy are similar to the changes shown by the mutation strategy learned by OB; that is, the exploration is emphasized first, and then the exploitation is biased.
> > > > However, traditional EA does not consider these. As the iteration continues, exploration and exploitation are considered equally. This is one of the reasons why BOptformer is better than EA baselines.
> > > >
> > > > Figure 3 shows the mutation strategy learned by the 5 OBs (actually FM) of BOptformer. The mutation strategy of OB1 tends to explore a broad solution space, and the mutation strategies of the next few OBs gradually shift from searching the vast space to searching the nearby space of the input population of OB.
> > > >
> > > > **sample-efficient**
> > > >
> > > > We're not sure if we get what you mean. If this part of the explanation is not clear, you are welcome to bring it up.
> > > >
> > > > The most significant advantage of BOptformer is sample efficiency; that is, BOptformer can find a better solution with a very small number of steps (No. of OBs). Because the learned optimization strategy in BOptformer fits the target function very well, it can guide sampling efficiently. Experimental results can illustrate the correctness of this statement. However, we admit that the sample efficiency of BOptformer is based on the high-fidelity function set. We hope to address this issue in the future.
> > > >
> > > > **Q2**: Does the pre-training phase require domain knowledge of the black-box we want to optimise? If so what is this knowledge and what are those assumptions?
> > > >
> > > > **Answer2** : BOptformer aims to establish a learned optimization strategy to match the target task. Therefore, in the training stage of BOptformer, we need the knowledge of the target task. We use the training set to acquire knowledge of the target task. The training set can contain surrogate functions of different fidelity for the target task. The high-fidelity function set contains more information about the target function, and the BOptformer trained on them has a stronger ability to solve the objective function; and vice versa. Of course, we admit that obtaining high-fidelity surrogate functions is challenging, and this is also a shortcoming of BOptformer. We can use UCBs and neural networks to build high-fidelity surrogate functions of black-box functions. The BOptformer trained on the low-fidelity function set still has good performance.
> > > >
> > > > For example, we train BOptformer on F1-F3 and make it successful on F4-F9.
> > > >
> > > > **From the expression of the functions**, F1-F3 are low-fidelity surrogate functions of the target functions F4-F9. F1-F3 contain the information of objective functions (F4-F9). For example, F7 can be decomposed into $\sum \limits_{i=1}^{D} z_i^2-\sum \limits_{i=1}^{D}10\cos(2\pi z_i)+ \sum \limits_{i=1}^{D}10, z_i=x_i-b_i$. F2 is the low-fidelity surrogate function of $\sum \limits_{i=1}^{D} z_i^2$. F1 is the low-fidelity surrogate function of $\sum \limits_{i=1}^{D}10\cos(2\pi z_i)$. For other functions in F4-F9, we can find similar surrogate functions from F1-F3. BOptformer can use this information to maximize the matching degree between the learned optimization strategy and the objective function. However, F6 is less similar to F1-F3 than F4, F5, and F7-F9. Therefore, although the performance of BOptformer on F6 is better than that of the comparison algorithm, it is still poor.
> > > >
> > > > **From the perspective of landscape features**, F1-F3 include the following features: unimodal, multimodal, separable, and non-separable. The landscape features included in F4-F9 are as follows:
> > > >
> > > > F4: Unimodal, Separable
> > > >
> > > > F5: Unimodal, Separable
> > > >
> > > > F6: Multimodal, Non-separable, **Having a very narrow valley from local optimum to global optimum**
> > > >
> > > > F7: Multimodal, Separable, **Asymmetrical, Local optima’s number is huge**
> > > >
> > > > F8: Multi-modal, Non-separable, **Rotated**
> > > >
> > > > F9: Multi-modal, Non-separable, **Asymmetrical**
> > > >
> > > > The landscape features of F4 and F5 can be found in F1-F3. F6-F9 all have new features. The interference strength of different characteristics to landscape is arranged as follows: **Having a very narrow valley from local optimum to global optimum**>**Asymmetrical, Local optima’s number is huge**>**Asymmetrical**>**Rotated**. Therefore, BOptformer has the best generalization performance on F4, F5, and F8, the second-best generalization performance on F7 and F9, and the worst on F6.

---

> > > > > ### Comment · Reviewer_YunP · 2022-12-11
> > > > > **Thank you**
> > > > >
> > > > > Thank you for the clarifications. What I meant by sample efficiency was how many samples we need to pre train the model. If we give this extra number of samples in the target domain, would standard black box optimisation do well? That is not to say this is a point against this paper. I am just wondering how black box optimisation fairs with the same number of data.

---

> > > > > > ### Author Response · Authors · 2022-12-11
> > > > > > **Reponse**
> > > > > >
> > > > > > Good question.
> > > > > >
> > > > > > If we say that we use the BOptformer training set for standard black-box optimization, how should they be used? In other words, how can they use this information to aid in the design of their algorithms? We can imagine that they can only use these to choose suitable hyperparameters and components themselves. A lot of work is also based on the author's wisdom to redesign new parts to suit the target task. Given the information on the target function, standard black-box optimization can only use a very small part of it to assist in the design of its own optimization strategy. **However, BOptformer is more intelligent and smarter, and it has taken a big step forward in efficiently using objective function information to assist in the design of optimization strategies.** This is also our primary motivation and contribution. Compared to the current method, it parameterizes all parts of the EA. Given some information about the target task, we can adjust these parameters to suit the target task.
> > > > > >
> > > > > > **For all cases in this paper, our experimental setup is unfair to BOptformer.** We use target functions to select appropriate hyperparameters and components for the compared algorithms. For example, we choose the optimal combination of crossover, mutation, and selection operators and their parameters for the genetic algorithm. BOptformer only relies on **low-fidelity surrogate functions of the target function** to aid in designing its optimization strategy. However, BOptformer outperforms the compared algorithms. Thus, BOptformer is sample-efficient.
> > > > > >
> > > > > > We also visualize the learned crossover and mutation strategies. Their behaviors confirm that the learned optimization strategies are efficient (see Section 4.5).
> > > > > >
> > > > > > **We also added more experiments to visualize the learned mutation strategy.**
> > > > > >
> > > > > > We visualized the mutation strategy of BOptformer to explore its behavior. We use polynomial mutation, which is commonly used in evolutionary algorithms, as a reference. Given the input population (input), the mutated population (OB1) is obtained through OB1; the new population (mutpolyn) is obtained by performing polynomial mutation on the input population. We visualized F4-F9. Sorry, we can no longer modify the original manuscript. The visualized results can be downloaded from the anonymous link below.
> > > > > >
> > > > > > https://anonymous.4open.science/r/ICLR-2023-Rebuttal-B753/vis_landscape.pdf
> > > > > >
> > > > > > We can observe the following phenomena:
> > > > > >
> > > > > > 1) The population generated by performing polynomial mutation is more evenly distributed on the landscape. However, most of the solutions produced by BOptformer's mutation strategy are concentrated in "areas with greater potential", which are closer to the optimal solution. Moreover, the population distribution generated by our scheme also takes diversity into account. In the non-optimal solution area, it is also more comprehensive than that of polynomial mutation, which is more conducive to jumping out of the local solution.
> > > > > >
> > > > > > 2) The population produced by performing polynomial mutation moves slightly compared to the original population. However, the mutation strategy of BOptformer can guide the input population to make big moves toward the optimal solution, which significantly accelerates the convergence of the algorithm.
> > > > > >
> > > > > > This shows that BOptformer is able to use the information of the objective function to guide the design of the mutation strategy, making it more applicable to the target optimization task, which is consistent with our motivation.

---

### Public Comment · ~Wenlong_Lyu1 · 2022-11-07
**Two ICLR'23 submissions that are very similar**

Somehow I found these two ICLR submissions very similar, with regard to the figures, formulae, and proposed methods in the two papers, the difference seems to be the AI models used (Transformer vs CNN)

1. [OPTFORMER: BEYOND TRANSFORMER FOR BLACKBOX OPTIMIZATION](https://openreview.net/pdf?id=sP0p5S-gZ2)
2. [DECN: EVOLUTION INSPIRED DEEP CONVOLUTION NETWORK FOR BLACK-BOX OPTIMIZATION](https://openreview.net/pdf?id=Ur_qORZ6-9R)

For example:
- Figure 1 of [1] and [2] are basically the same
- Figure 2 of [1] and figure 6 for [2] are basically same
- Eq. 7 of [2] and Eq. 12 of [1] are the same
- Algorithm 1 of both [1] and [2] are very similar

---

> ### Author Response · Authors · 2022-11-18
> **Response to Wenlong Lyu**
>
> First of all, thank you for carefully reviewing our article. We also invite you to re-evaluate our paper. In fact, both of these articles are our work. As for the similarities between the two articles you mentioned, we make the following explanations:
>
> **Q1**: Figure 1 of [1] and [2] are basically the same.
>
> **Answer 1:** These two diagrams are intended to show that we propose an end-to-end optimization framework without referring to the specific model architecture, parameter settings, and other information. Moreover, we redrew Figure 1 of [1]. Please check our revised version.
>
> **Q2**:  Figure 2 of [1] and figure 6 for [2] are basically the same.
>
> **Answer 2:** The style of these two pictures is similar, but the content they express is entirely different. In order to avoid misunderstanding, we have redrawn Figure 2 of [1].
>
> **Q3**: Eq. 7 of [2] and Eq. 12 of [1] are the same.
>
> **Answer 3:** This equation represents the loss function used in model optimization. BOptformer uses the loss function designed by DECN, and we have added a reference to DECN.
>
> **Q4**: Algorithm 1 of both [1] and [2] are very similar.
>
> **Answer 4:** This algorithm presents the method of parameter updating. We believe that there is no problem with the similarity between [1] and [2] in the method of parameter updating, and it will not affect the contribution stated in the article. We have added the reference of DECN in the modified version.
>
> BOptformer is a new optimization framework designed by modifying the MSA module, FFN module, and residual structure employed in Transformer. DECN is a new optimization framework proposed by us. DECN is not a variant of CNN. We just modified the convolution operation in the CRM module to achieve the purpose of information interaction between individuals. Its architecture and modules are a completely new design. For the L2O community, we design two competitive frameworks for black-box optimization.
>
> Finally, thanks again for your comments.

---

### Decision · Program_Chairs · 2023-01-20

**Decision:**

Reject

**Justification For Why Not Higher Score:**

The main concerns of clarification and lack of thorough empirical evaluation are not fully addressed after the review-author discussion.

**Justification For Why Not Lower Score:**

N/A

**Metareview: Summary, Strengths And Weaknesses:**

This paper proposes a new network architecture based on transformers to learn an evolutionary algorithm for black-box optimization. It modifies the structure of transformers and proposes new components for crossover, mutation and selection. By training on surrogate functions similar to the target test function, the paper shows better convergence than a few standard EA baselines.

Strengths:
- A novel architecture for learning EA operations.
- Strong performance than baselines on the problems evaluated

Weaknesses:
- English writing and presentation could be improved. The authors' feedback clarifies many questions from the authors. A significant amount of revision may be required to improve the quality of writing.
- Multiple reviewers have concerns on the set of experiments. Only two real-data experiments are conducted in addition to synthetic test functions.
- More visual ablations would help readers understand the behavior of the learned model.

**Summary Of Ac-Reviewer Meeting:**

N/A